**On the potential of Cluster Ion Counter (CIC) to observe local new particle formation, condensation sink and growth rate of newly formed particles**

Markku Kulmala[1,2,3], Santeri Tuovinen[1], Sander Mirme[4,5], Paap Koemets[4,5], Lauri Ahonen[1], Yongchun Liu[2], Heikki Junninen[.4], Tuukka Petäjä[1,2,3] and Veli-Matti Kerminen[1]

[1] Institute for Atmospheric and Earth System Research (INAR)/Physics, University of Helsinki, Helsinki, 00014, Finland
[2] Aerosol and Haze Laboratory, Beijing Advanced Innovation Center for Soft Matter Science and Engineering, Beijing University of Chemical Technology, Beijing, 100089, China
[3] Joint International Research Laboratory of Atmospheric and Earth System Sciences, School of Atmospheric Sciences, Nanjing University, Nanjing, 210023, China
[4]Institute for Physics, University of Tartu, Tartu, 50090, Estonia
[5]Airel Ltd., Observatooriumi 5, 61602 Tõravere, Estonia

Keywords: atmospheric ions, ion measurements, cluster ions, intermediate ions, instrumentation, new particle formation, condensation sink

*Correspondence to*: Markku Kulmala (markku.kulmala@helsinki.fi)

**Abstract**

Cluster Ion Counter (CIC) is a simple 3-channel instrument designed to observe ions in the electrical mobility equivalent diameter range from 1.0 to 5 nm. With the three channels, we can observe concentrations of both ion clusters (sub-2 nm ions) and intermediate ions. Furthermore, as derived here, we can estimate condensation sink (CS), intensity of local new particle formation, growth rate of newly formed particles from 2 nm to 3 nm, and formation rate of 2 nm ions. We compared CIC measurements with those of a multichannel ion spectrometer, the Neutral cluster and Air Ion Spectrometer (NAIS), and found that the concentrations agreed well between the two instruments, with the correlation coefficients of 0.89 and 0.86 for sub-2 nm and 2.0–2.3 nm ions, respectively. According to the observations made in Hyytiälä, Finland and Beijing, China, the ion source rate was estimated to be about 2–4 ion pairs $cm^{-3}$ $s^{-1}$. The new CIC is a simple and cheap instrument that can be used in different environments to obtain information about small ion dynamics, local intermediate ion formation and CS in a robust way when combined with the theoretical framework presented here.

**1. Introduction**

New particle formation (NPF) is the dominant source of the number concentration of aerosol particles in the global atmosphere (Gordon et al., 2017), thereby having potentially large influences on global climate (e.g. Boucher et al., 2013) and regional air quality (e.g. Guo et al., 2014; Kulmala et al., 2022). During the past 2-3 decades, atmospheric NPF has been characterized in terms of the particle formation and growth rates at a vast variety of sites in different atmospheric environments (Wang et al., 2017; Kerminen et al., 2018; Nieminen et al., 2018; Chu et al., 2019; Bousiotis et al., 2021). Such characteristics describe mainly regional NPF, i.e. NPF averaged over relatively large spatial scales of at least tens of km. Much less information is available about local NPF, or about the small-scale variability of

regional NPF (Kulmala et al., 2024a, 2024b). Such information would be important in
identifying hot spot areas for atmospheric NPF or estimating the relative importance of
various local sources to regional NPF.
Atmospheric cluster ion (diameters below 2 nm) measurements can provide insight into ion
source processes, such as the ion production rate associated with different atmospheric
ionization pathways, as well as ion loss processes, such as ion-ion recombination or
scavenging of ions by a pre-existing atmospheric aerosol population (e.g. Hirsikko et al.,
2011; Kontkanen et al., 2013). Observations of intermediate ions (diameters between 2 and 7
nm) can be used to get information about atmospheric NPF (e.g. Tammet et al., 2014),
whereas small intermediate ions (approx. 2.0–2.3 nm) can be used to detect "local" NPF, i.e.
NPF taking place within a close proximity of a measurement site (Tuovinen et al., 2024).
Intermediate ions are sensitive to both occurrence and intensity of atmospheric NPF (e.g.
Horrak et al., 1998; Tammet et al., 2014, Leino et al., 2016). Recently, Kulmala et al. (2024a)
and Tuovinen et al. (2024) found that the smallest sizes of intermediate ions describe
relatively well the local production of new aerosol particles. These results were obtained
using a Neutral Cluster and Air Ion Spectrometer (NAIS; Mirme and Mirme, 2013). The
NAIS is, however, a sophisticated instrument that provides information not necessarily
needed when investigating local NPF, such as detailed knowledge of both ion and particle
number size distributions.
In this study, we will analyze data obtained using a Cluster Ion Counter (CIC; Mirme et al.,
2024), a recently developed and simple 3-channel instrument, and will investigate how this
instrument can be utilized to determine several variables important to NPF and small ion
dynamics. Our main objectives are to derive simple equations for characterizing ion
dynamics related to local NPF, and to find out whether the CIC is sensitive and reliable
enough for such purposes. In order to reach these objectives, we will first derive equations
that can be used to estimate condensation sink (CS), growth rate of newly formed particles
and formation rate of 2 nm ions, quantifying the intensity of local new particle formation
(actually local intermediate ion formation, LIIF), based on CIC measurements. Next, we will
compare ion concentrations between the CIC and NAIS, as measured at the SMEAR II
station in Hyytiälä, Finland. Finally, we will demonstrate how to apply CIC measurements in
practice for obtaining information about local NPF and related quantities, including the
condensation sink.
**2. Material and Methods**
**2.1 Cluster Ion Counter (CIC)**
The Cluster Ion Counter (CIC) is an instrument for measuring the total number concentration
of both positive and negative cluster ions. The CIC uses two separate first-order cylindrical
differential mobility analyzers, one for each polarity (Tammet, 1970). The principal
components of the analyzers are a central electrode on the axis of the analyzer that is held at a
steady voltage, and three cylindrical collecting electrodes flush with the outer wall of the
analyzer which are at zero electric potential. A constant sample flow is produced through the
analyzer using a blower at the outlet. The sampled ions passing through the analyzers are
repelled by the central electrode and they may deposit on one of the collecting electrodes
depending on the electrical mobility of the ions. The electric current produced by the
deposited ions is measured using high precision integrating electrometers (Mirme et al.,
102  2024).
The mobility dependent detection efficiency curves of the three channels are determined by
the geometry of the analyzer, sample air flow rate and the electric voltage of the central
electrode. According to the idealized model of differential mobility analyzers (Tammet,
1970), the primary parameters governing the detection efficiency curves and the limiting
mobilities of the collecting electrodes are the electrical capacitances between the central
electrode and the each collecting electrode, as well as the ratio of sample flow rate to central
electrode voltage.  The original CIC was designed to allow the estimation of average cluster
ion mobility. However, the device can easily be modified to focus on other aspects of the
mobility distribution.
In the CIC, the flow rate-to-voltage ratio can be freely adjusted through software. The lengths
of the collecting electrodes and geometry of the central electrode of the CIC can be changed
without requiring additional modifications to the device.
A modified analyzer for the CIC was developed to estimate the concentration of intermediate
ions roughly between 2.0 and 2.3 nm. Due to the relatively simple construction of the CIC,
and specifically the absence of a separate sheath air flow layer in the mobility analyzer, the
detection efficiency curves of the individual electrodes of the CIC are relatively wide and
extend far towards larger particles (Figure 1). However it is notable that for particles beyond
certain size the transfer functions differ only by a constant coefficient. We can use the signal
from one channel to compensate for the concentration of larger particles in another channel
and virtually achieve a higher size resolution.
We altered the collecting and central electrode geometry, as well as voltage, and flow rate
within the mechanical constraints of the original device so that the transfer functions of
channel 2 and 3 would differ only in a relative narrow size range and the difference would
peak between 2.0 and 2.3 nm. This required extending the first collecting electrode and
shortening the second and third electrode, as well as changing the diameter and length of the
central electrode.
In the modified CIC, the signal from the first electrometer can be used to estimate the cluster
ion concentrations. By subtracting the signal of the third channel from the signal of the
second channel, the concentration of intermediate ions roughly between 2.0 and 2.3 nm can
be estimated, denoted by Channel 2-3 from now on. The third channel can be utilized for ions
from 2.3 to 5 nm.
**2.2 Theoretical framework**
The time evolution of sub-2 nm ion concentration, $I$, can be written as
$$\frac{dI}{dt} = Q - \alpha I^2 - \text{CoagS}_I \times I, \hspace{3cm} (1)$$
where $Q$ is the ion source rate, $\alpha$ ($\approx 1.6 \times 10^{-6}$ cm$^3$ s$^{-1}$; Franchin et al., 2015) is the ion-ion
recombination rate, and CoagS$_I$ is the coagulation sink of the sub-2 nm ions onto pre-existing
aerosol particles. Other losses, such as deposition are assumed to be negligible. In a pseudo-
steady state, we may approximate the left-hand side of eq.1 equal to zero, from which we
obtain:

$$\mathrm{CoagS_I} = Q/I - \alpha I. \qquad (2)$$


The coagulation sink of neutral particles of diameter $d_p$ can be connected with the
condensation sink (CS) of sulphuric acid monomers via (see Lehtinen et al., 2007)

$$\mathrm{CS} \approx \mathrm{CoagS}(d_p)\,(d_p/0.7\ \mathrm{nm})^m\,, \qquad (3)$$


where the exponent $m$ depends on the shape of the pre-existing particle number size
distribution, and the diameter of a sulphuric acid monomer is estimated to be 0.7 nm. By
combining eqs. 2 and 3 we then obtain:

$$\mathrm{CS} \approx \mathrm{CoagS}(d_p = d_{p,I}) \times [d_p/0.7\ \mathrm{nm}]^m \times [Q/I - \alpha\,I\,] / \mathrm{CoagS_I}\,, \qquad (4)$$


where $d_{p,I}$ refers to the median diameter of the sub-2 nm ions. In order to simplify eq. 4, we
will make three further approximations: 1) $d_{p,I}$ is equal to 1.2 nm for negative cluster ions
observed with CIC, and 1.0 nm for negative cluster ions measured with NAIS, 2) the
exponent $m$ is equal to 1.6 (see Lehtinen et al., 2007), and 3) the ratio $\mathrm{CoagS}(d_p = d_{p,I})$ /
$\mathrm{CoagS_I}$ is equal to 0.5 (Leppä et al., 2011; Mahfourz and Donahue, 2021). The $d_{p,I}$ were
determined as weighted mean diameters of 0.8-2.0 nm (NAIS) and 1.0-2.0 nm (CIC) negative
ions based on the NAIS ion number size distributions. The concentrations of ions in different
size bins were used as weights. By combining these approximations, we finally obtain:

$$\mathrm{CS} \approx 1.2\,(Q/I - \alpha\,I). \qquad (5a)$$


$$\mathrm{CS} \approx 0.9\,(Q/I - \alpha\,I). \qquad (5b)$$


Here we utilize eq. 5a if $I$ is measured with the CIC and eq. 5b if $I$ is measured with the
NAIS.

Similar to eq. 1, the time evolution of the concentration of the smallest (2.0–2.3 nm)
intermediate ions, $N$, can be written as

$$\frac{dN}{dt} = J_2 - \mathrm{CoagS_N} \times N - J_{\mathrm{out}}, \qquad (6)$$


where $J_2$ is the formation rate of 2 nm ions, $\mathrm{CoagS_N}$ is the coagulation sink of the 2.0–2.3 nm
ions onto the pre-existing aerosol population, and $J_{\mathrm{out}}$ is the rate at which these ions grow out
of the 2.0–2.3 nm size range. $\mathrm{CoagS_N}$ and $J_{\mathrm{out}}$ can be approximated as:

$$\mathrm{CoagS_N} \approx \mathrm{CoagS_I} \times (1.2\ \mathrm{nm}\,/2.1\ \mathrm{nm})^{1.6} \approx 0.4\ \mathrm{CoagS_I} \approx 0.4\,(Q/I - \alpha I), \qquad (7)$$


$$J_{\mathrm{out}} \approx \mathrm{GR_{2.3\,nm}} \times N/\Delta\mathrm{d}, \qquad (8)$$


where $\mathrm{GR_{2.3\,nm}}$ is the growth rate of 2.3 nm ions and $\Delta\mathrm{d}$ (=0.3 nm) is the width of the
intermediate ion channel of the CIC. Assuming a pseudo-steady state (d$N$/d$t$ = 0) and using
Eqs. 2, 7 and 8, we then obtain:

$$J_2 = 0.4\,(Q/I - \alpha I) \times N + \mathrm{GR_{2.3\,nm}} \times N/\Delta\mathrm{d} + \alpha IN. \qquad (9)$$


The last term in Eq. 9 accounts for the loss rate of 2.0-2.3 nm ions due to their recombination
with sub-2 nm ions.
Particle (or ion) growth rates can be determined from the following equation:
$$GR = \frac{\Delta d_i}{\Delta t},$$ (10)
where $\Delta d_i$ is the change of the diameter of ions over the time interval $\Delta t$ as the ions grow in
size. In section 3.2 we will demonstrate how the CIC measurement can be used for
determining growth rates.
**2.3. Observations and data**
The CIC and NAIS were compared with each other at the SMEAR II station in Hyytiälä (Hari
and Kulmala, 2005) during 16 January–01 April, 2024; however, NAIS data were missing
from the period 16-17 March. The NAIS (Neutral Cluster and Air Ion Spectrometer) is a
multichannel instrument to measure atmospheric ions from 0.8 to 42 nm and total particle
concentrations from 2.5 to 42 nm (Mirme and Mirme, 2013). From NAIS, concentrations of
total sub-2 nm ions, 1-2 nm, and 2.0-2.3 nm were used in this study. In addition, as CIC
Channel 2-3 covers a slightly wider diameter range than 2-2.3 nm, we determined
concentrations corresponding to those within the same mobility diameter range from the ion
number size distributions measured by NAIS (NAIS Channel 2-3). The NAIS ion number
size distributions were multiplied by the detection efficiencies for the CIC Channel 2–3
(Figure 1), and then summed. The resulting total concentrations were assumed to correspond
to the detected ion concentration by CIC Channel 2-3. This concentration was then divided
by the average detection efficiency for the CIC Channel 2-3 to get the atmospheric ion
concentration. If the NAIS concentrations are assumed to be equal to the atmospheric
concentrations, then in theory the CIC and NAIS Channel 2-3 concentrations should be equal.
For convenience, CIC Channel 2-3, NAIS 2.0-2.3 nm, and NAIS Channel 2-3 are collectively
referred to as 2.0-2.3 nm ions when separating them is not necessary.
Furthermore, the conceptual model (see chapter 2.2) was used to analyse the data from both
SMEAR II and AHL/BUCT station in Beijing, China (Liu et al., 2020). In data analysis we
use 10%, 25%, 50%, 75%, and 90% percentiles for small and intermediate ion
concentrations and CS values. A longer time spans were used for this part of the analysis. For
Hyytiälä, the data cover most of the time between the beginning of 2016 and end of 2020. For
Beijing, ion concentrations were determined over the period 13 January 2018 to 01 April
2020, whereas the CS data cover the period 20 February 2018 to 31 March 2019. The particle
number size distributions to derive the CS data were measured by a twin DMPS (Differential
Mobility Particle Sizer; Aalto et al., 2001) in Hyytiälä and in Beijing by a particle size
distribution (PSD) system (Liu et al., 2016). See Zhou et al. (2020) for more details on the
measurements in Beijing.
**3. Results and Discussion**
**3.1 Instrument comparison**
In order to find out how reliably the CIC is able to observe ion concentrations, we compared
it with the NAIS at the SMEAR II station in Hyytiälä, Finland. Tables 1 and 2 summarize the
percentiles of the ion concentrations measured by these two instruments for different size
fractions. We can see that the total concentration of sub-2 nm negative ions measured by the
NAIS is significantly higher than those measured by the CIC (channel 1), the median
concentrations being equal to 530 and 210 cm$^{-3}$, respectively. This result is expected, as the
detection efficiency of both instruments decreases rapidly for particles smaller than 1 nm.
However, the NAIS is able to correct for this in data inversion, while the CIC is not due to
the lack of detailed information about the measured size distribution. Excluding the smallest
ions measured by the NAIS, i.e. considering only the 1–2 nm size range, the median
concentration drops down to 180 cm$^{-3}$. This is slightly below the median sub-2 nm
concentration measured by the CIC, but only about one third of the median total sub-2 nm ion
concentration measured by the NAIS.
A comparison between the two instruments is in Figure 2 for small (1-2 nm) ions, and in
Figure 3 for the smallest size class of intermediate ions (2.0–2.3 nm).  We can see that when
the small ion concentration is above 200 cm$^{-3}$, the two instruments show similar values, while
at lower concentrations there is more spread in the values with the CIC generally measuring
higher concentrations than the NAIS. At low concentrations, it is possible that the
uncertainties in the detection efficiencies of the ions with diameters close to 1 nm impact the
results, explaining our observations. CIC Channel 2-3 concentration are consistently lower
than NAIS Channel 2-3 concentrations, with the difference being smaller when the
concentrations are higher, suggesting that a lower concentrations electronic noise impacts the
comparison increasingly. There is more spread between the values of NAIS 2.0-2.3 nm and
CIC Channel 2-3. At higher concentrations the CIC shows higher concentrations than NAIS
2.0-2.3 nm concentration. However, the overall the overall agreement between these two
instruments is good with the correlation coefficients of 0.85 and 0.86 for small ions and 2.0-
2.3 nm ions, respectively.
Figure 4 presents the time series of ion concentrations measured by the CIC and NAIS over
the whole two and half-month period, while Figure 5 presents the diurnal pattern of ion
concentrations on a selected day (10[th] of March, 2024). Total sub-2 nm ion concentrations
measured by the NAIS are higher than CIC Channel 1 ion concentrations. However, for
majority of the time (see Figure 4), the NAIS 1-2 nm ion concentration and CIC Channel 1
concentration are close to each other. On the selected day, CIC Channel 2-3 peak NAIS
Channel 2-3 values are similar, 60 and 80 cm$^{-3}$, respectively, whereas the NAIS 2.0-2.3 nm
peak value is lower at around 20 cm$^{-3}$. CIC Channel 2-3 is likely influenced by ions larger
than 2.3 nm, impacting the measured concentration when intermediate ion concentration is
high, such as during NPF. The correlation coefficient between the concentrations from the
two instruments on the selected day is around 0.9 for both sub-2 nm and 2.0-2.3 nm ions.
Comparing the lower percentiles in Tables 1 and 2, it is apparent that a large fraction of CIC
Channel 2-3 concentrations are negative. At very low concentrations (< 1 cm$^{-3}$), the signal is
mainly noise. In addition, Figure 4 and 5 show that the low background concentrations
measured by CIC Channel 2-3 are on average less than 10% of NAIS Channel 2-3
concentrations, which we postulate is due to estimation errors caused by the limited size
resolution of the NAIS as well as different background noise levels of the instruments . At
very low concentrations, the values from either instrument can be considered unreliable.
Regardless, within the scope of this study, these background concentrations are of less
interest compared to the higher concentrations. Periods of LIIF can be identified based on
elevated 2.0-2.3 nm ion concentrations, and these ion concentrations can then be used to
derive parameters, such as the ion formation rate, to quantify the intensity of LIIF. The

comparison of the two instruments done here has shown that we can use CIC measurements to identify LIIF.

**3.2 Application of CIC measurement in investigating condensation sink and local NPF**

Figure 6 illustrates how the estimated condensation sink (CS) based on Eq. 5b behaves as a function of small ion concentrations, $I$, for different ion production rates. In the same plot, we have included the observed variability of CS as determined from the particle number size distributions and $I$ in both Hyytiälä and Beijing. We can see that measured and theoretically calculated estimates of CS agree with each other the best when median ion production rates are between about 2 and 4 ion pairs $cm^{-3}$ $s^{-1}$ in both Hyytiälä and Beijing.

The CIC has a higher detection efficiency for small ions than the NAIS due to a shorter inlet tract, and consequently, lower inlet losses. However, in case of both instruments, the detection efficiency for sub-2 nm ions is very strongly dependent on particle size. The NAIS measures the size distribution of ions, and the data inversion algorithm uses that information to correct for the size-dependent detection efficiency. The CIC has limited information about the size distribution of detected ions, making it more difficult to correct for the detection efficiency. Using the sub-2 nm ion concentrations from the NAIS and the CIC (Tables 1 and 2), we estimated how the concentrations measured using the CIC and NAIS will influence the estimated values of CS. By using eq. 5 and by assuming the median sub-2 nm ion concentrations measured by these two instruments (Tables 1 and 2), we may calculate that the values of CS measured using the NAIS are 0.237, 0.256 and 0.266 times those measured using the CIC for $Q$ equal to 2,3 and 4, respectively. Therefore, if we use the CIC for estimating CS via eq. 5a, the real CS (using NAIS and equation 5b) is about 0.25 times the one observed by CIC.

Figure 7 shows the CS derived based on Eq. 5a and 5b versus CS determined from the full particle number size distribution ($CS_{DMPS}$). We see that the CS predicted by NAIS varies less than $CS_{DMPS}$, but is mostly within the same order of magnitude. CS predicted by CIC is consistently higher than $CS_{DMPS}$. However, considering the above discussion, and multiplying the estimated CS by 0.25, we get values much closer to $CS_{DMPS}$. Assuming Q=2, the CS values predicted by CIC are mainly within a factor of three from $CS_{DMPS}$ values.

We have assumed that the only losses of ions are due to their coagulation with larger particles and their recombination with oppositely charged ions. In reality, processes such as deposition also affect the ion concentration. For example, Tammet et al. (2006) found that in Hyytiälä deposition of ions to forest canopy impacts small ion concentrations. In addition, we have assumed the ion source rate to be constant. In reality, it is expected to vary somewhat, for example due to varying radon concentration (e.g., Hirsikko et al., 2007). Therefore, the presented method of determining CS can only give a rough approximation for CS.

In order to illustrate how the CIC can be used to determine the ion growth rate (GR), we selected one measurement day (Figure 8) and determined GR using the appearance time method (e.g. Lehtipalo et al., 2014) and equation (10). Ion concentrations from the CIC Channel 2-3 and Channel 3 from February 13[th] were used. The ion concentrations were smoothed using a moving 1-hour median method to lessen the impact of noise. As we can see from Figure 8, Channel 3 and Channel 2–3 concentrations on the selected day have a similar shape between 10:00 and 16:00, and the shape of the Channel 3 roughly follows that of Channel 2–3 with a time delay. Considering the shape and features of the two curves, and the

times at which the two concentrations reach a similar fraction of the maximum concentration
(appearance time method,), two time instances were identified visually. The appearance times
were chosen to correspond to times when the ion concentrations were around 20 % of the
maximum concentrations. From these approximate appearance times, a time delay was
calculated. Based on Figure 1, the diameters of 2.2 nm and 2.9 nm for Channel 2–3 and 3
were assumed, respectively. We note that on this particular example day, the curves follow
each other closely for a span of several hours, so that the value of GR is not very sensitive to
the identified appearance times, i.e., the chosen fraction of the maximum concentration
anywhere between 0.2 and 0.9 results in the same approximate GR. The resulting GR was
approximately 0.9 nm/h. This value is in the expected range, as the earlier long-term
measurements at the same site indicate typical growth rates between about 1 and 2 nm/h for
sub-3 nm ions (Hirsikko et al., 2005; Manninen et al., 2010). We should note, however, that it
is not possible to determine GRs for all measurement days using the procedure presented
here. This is because even if an increase in ion concentrations was observed, the signal might
be too noisy, making the determination of appearance times too unreliable. In addition, not all
days exhibited a clear delay between the two appearance times, making the determination of
growth rate impossible.
Using Eq. 9, we can estimate the formation rate of 2 nm ions, $J_2$. Figure 8 shows these
formation rates for Hyytiälä and Beijing. This formation rate can be given as a function of the
measured number concentrations of 2.0–2.3 nm intermediate ions, in addition to which $J_2$
depends on the growth rate, ion source rate, and ion loss rate, the latter of which was
estimated using the sub-2 nm ion concentrations according to Eq. 5b. $J_2$ also depends on the
concentration of sub-2 nm ions, which is determined by the ion loss rate and ion source rate
(Eq. 1). For Figure 9, the median sub-2 nm ion concentrations in Hyytiälä and in Beijing
were used in Eq. 9. The most probable values are 1–2 nm/h for the growth rate in Hyytiälä
(Figure 8,  Hirsikko et al., 2005; Manninen et al., 2010), 1–3 nm/h for the growth rate in
Beijing (Deng et al., 2020), and 2–3 $cm^{-3}$ $s^{-1}$ for the ion source rate (Figure 6). However, also
higher values are given for comparison. Manninen et al. (2010) calculated a median value of
0.06 $cm^{-3}$ $s^{-1}$ for $J_2$ based on long-term measurements in Hyytiälä, which is at the higher end
of values estimated in Figure 9. Compared with Hyytiälä, we estimate a factor of 2–4 larger
values of $J_2$ for Beijing. If one wants to estimate the total 2 nm particle formation rate, in both
places, it is considerably larger than the formation rate of 2 nm ions, being of the order of one
magnitude in Hyytiälä (Manninen et al., 2010, Kulmala et al., 2013) and even larger in
Beijing (Deng et al., 2020). These results are fully consistent with the general finding that on
average, observed new particle formation rates are 1 to 3 orders of magnitude larger in
polluted urban environments compared with clean or moderately polluted environments
(Kerminen et al., 2018; Nieminen et al., 2018), whereas the average formation rates of 2 nm
ions are typically within a factor of 2–3 between different environments (Manninen et al.,
390 2010).
Figure 10 shows the estimated time evolution of the condensation sink and 2-nm ion
formation rate during one day. The estimated value of CS varies only little, less than a factor
of 1.5, whereas the ion formation rate varies by more than two orders of magnitude during the
day. We can clearly see that when the estimated CS is at its lowest at around midday, the ion
formation rate is at its highest.
**4. Conclusions and summary**

The recent progress on finding local NPF (e.g. Kulmala et al., 2024; Tuovinen et al., 2024)
has opened a question: are we able to utilize a simple ion counter to identify and quantify
LIIF in a proper way? According to our results presented above, the answer is: yes.

We have developed a modified version of the CIC to measure sub-2 nm ion and 2.0–2.3 nm
ion concentrations as accurately as possible (Mirme et al., 2024). From the former quantity
we get information on the dynamics of small ions, including an estimate of the coagulation
sink of ions and, via equations (2) and (5), also condensation sink. Furthermore, the CIC
makes it possible to estimate the growth rate from about 2 nm to 3 nm and, with this
information, the formation rate of 2 nm ions, which we can use to quantify the intensity of
LIIF. While we have focused on negative ions in this paper, the same principles are also valid
for positive ions. LIIF is more sensitive to negative ions (Tuovinen et al., 2024), and thus
negative ions were investigated.

We compared the CIC with the NAIS in Hyytiälä, which demonstrates that the measured ion
concentrations from CIC are able to capture the temporal behavior of the ions such as the
variation in concentrations due to LIIF. The comparison of the estimated condensation sink
from ion concentrations using the ion balance equation with the observed ones in Hyytiälä
and Beijing demonstrates how the CIC, together with the simple theoretical framework, can
be used to estimate condensation sink, coagulation sink of ions and the ion formation rate. In
addition, the comparison of estimated CS based on CIC measurements with the CS
determined particle number size distributions shows that we can get estimates that are within
a factor three of the real CS. . Therefore, we can conclude that the CIC is an effective
instrument to observe LIIF and CS. Since CIC is ca seven times cheaper and requires less
maintenance than NAIS, with CIC one can have more observation locations and have wider
data coverage than with NAIS. However, if we want to investigate aerosol formation and
growth rates for the nucleation mode (3–25 nm), as is usually the case in investigating
regional NPF, NAIS measurements are needed.

**Author contribution**

Markku Kulmala had the original idea after discussions with Heikki Junninen. SM and PK
developed the CIC. LA performed CIC and NAIS comparison in Hyytiälä. ST and MK
analyzed the data. VMK and MK derived the used equations.  YL lead the observations in
Beijing and TP in Hyytiälä. HJ, VMK, TP and ST contributed to developing the idea further.
MK, VMK and ST  wrote the first version of the paper. All coauthors contributed the final
version of the paper.

**Competing interests**

Markku Kulmala is a member of the editorial board of Aerosol Research. The authors have
no other competing interests to declare.

**Acknowledgements**
We acknowledge the following projects: ACCC Flagship funded by the Academy of Finland
grant number 337549 (UH) and 337552 (FMI), Academy professorship funded by the Academy
of Finland  (grant no. 302958), Academy of Finland projects no. 1325656, 311932, 334792,
316114, 325647, 325681, 347782, "Quantifying carbon sink, CarbonSink+ and their

interaction with air quality" INAR project funded by Jane and Aatos Erkko Foundation,
"Gigacity" project funded by Wihuri foundation, European Research Council (ERC) project
ATM-GTP Contract No. 742206, European Union via Non-CO2 Forcers and their Climate,
Weather, Air Quality and Health Impacts (FOCI), and Estonian Research Council project
PRG71. University of Helsinki support via ACTRIS-HY is acknowledged. University of
Helsinki Doctoral Programme in Atmospheric Sciences and the High-End Foreign Expert
Recruitment Program of China (G2023106004L) is acknowledged. Support of the technical
and scientific staff in Hyytiälä SMEAR II station and AHL/BUCT station in Beijing are
acknowledged.

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

**Tables**

Table 1. Percentiles of the CIC Channel 1 (small ion) and Channel 2–3 (roughly 2.0–2.3 nm
ion) concentrations (cm$^{-3}$) during 16.01.2024–01.04.2024. Positive polarity is marked by +
and negative by –. The negative concentrations for the Channel 2 subtracted by Channel 3 are
indicative of a noisy signal of the instrument.

|  | Channel 1 | | Channel 2 - 3 | |
|---|---|---|---|---|
|  | + | – | + | – |
| Mean | 280 | 220 | 2.8 | 5.2 |
| 10% | 130 | 90 | −11 | -13 |
| 25% | 190 | 140 | −4.4 | -5.6 |
| 50% | 270 | 210 | 1.3 | 0.9 |
| 75% | 360 | 290 | 7.9 | 9.6 |
| 90% | 430 | 380 | 17 | 24 |

Table 2. Percentiles of NAIS concentrations (cm$^{-3}$) during 16.01.2024 – 01.04.2024,
excluding 16–17.03.2024. Small ions in the diameter ranges 0.8–2 nm and 1–2 nm are
included. Intermediate ion concentrations are included for diameter range 2.0–2.3 nm, as well
as for the diameter range that the CIC covers (Channel 2–3, see Sect. 2.3 for details).
Positive polarity is marked by + and negative by –.

|  | 0.8–2 nm | | 1–2 nm | | 2.0–2.3 nm | | Channel 2–3 | |
|---|---|---|---|---|---|---|---|---|
|  | + | – | + | – | + | – | + | – |
| Mean | 490 | 540 | 400 | 210 | 2.0 | 2.3 | 17 | 13 |
| 10% | 360 | 400 | 270 | 95 | 0.2 | 0.04 | 8.7 | 2.8 |
| 25% | 410 | 460 | 330 | 120 | 0.7 | 0.3 | 11 | 4.5 |
| 50% | 490 | 530 | 400 | 180 | 1.5 | 1.1 | 14 | 7.5 |
| 75% | 570 | 620 | 470 | 270 | 2.7 | 2.6 | 19 | 14 |
| 90% | 640 | 700 | 540 | 380 | 4.2 | 4.8 | 29 | 26 |


**Figures**

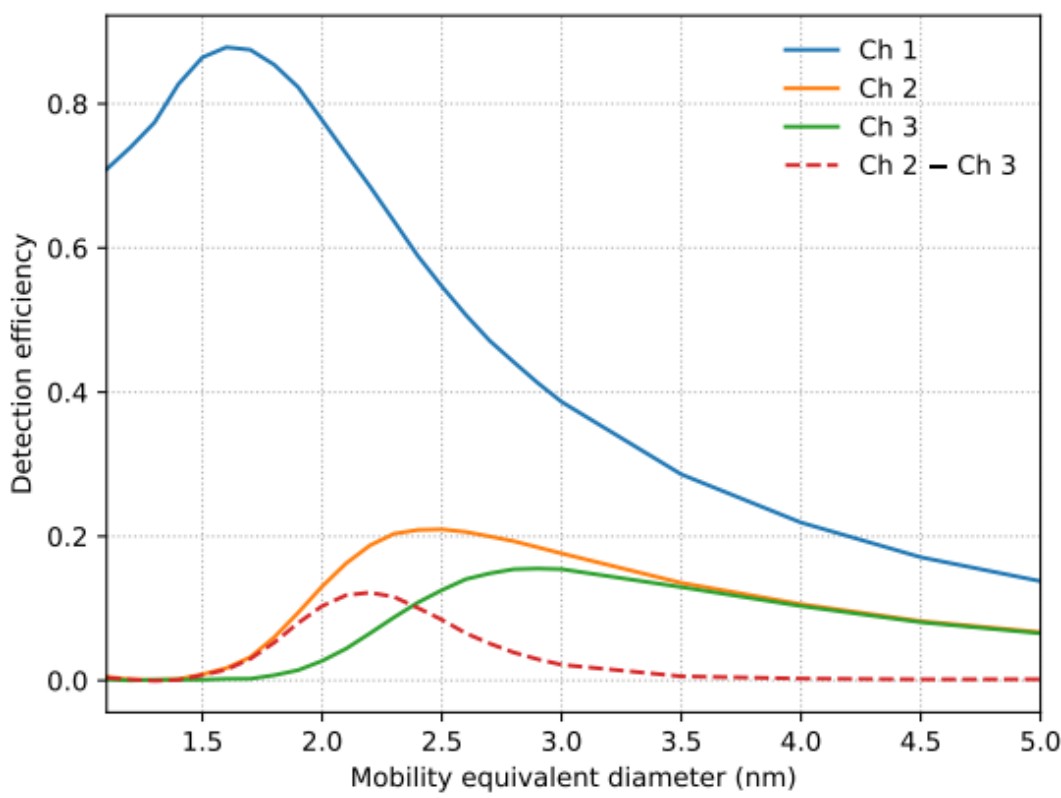

Figure 1. Experimental detection efficiency for ions in the range from 1.1 to 5.0 nm for each
of the 3 collecting electrodes of the CIC. Due to the absence of a separate sheath air flow
layer in the mobility analyzer, the detection efficiencies do not have a sharp upper size limit;
instead, they asymptotically approach zero as particle size increases. Ion concentrations in a
narrower size range can be estimated by subtracting the signal of channel 3 from channel
2. The detection efficiencies of the two channels converge from 2.5 nm to 3.5 nm and are
practically equal for larger particles.

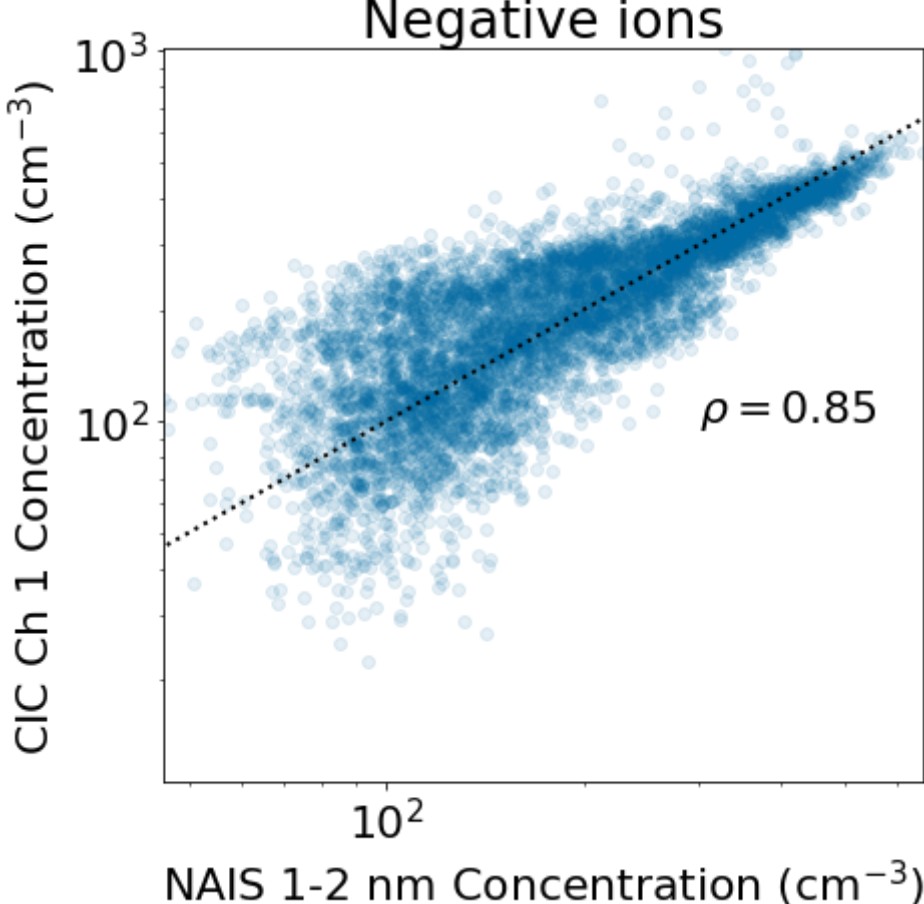

Figure 2. Scatter plot of the 15-min median negative small ion concentration measured with
the CIC as a function of the concentration measured with the NAIS in Hyytiälä. The NAIS
concentrations are from the diameter range 1–2 nm, while the CIC concentrations are from
Channel 1. The black dotted line marks the 1:1 line. Pearson correlation coefficient $\rho$ of the
two concentrations shown is included in the figure.

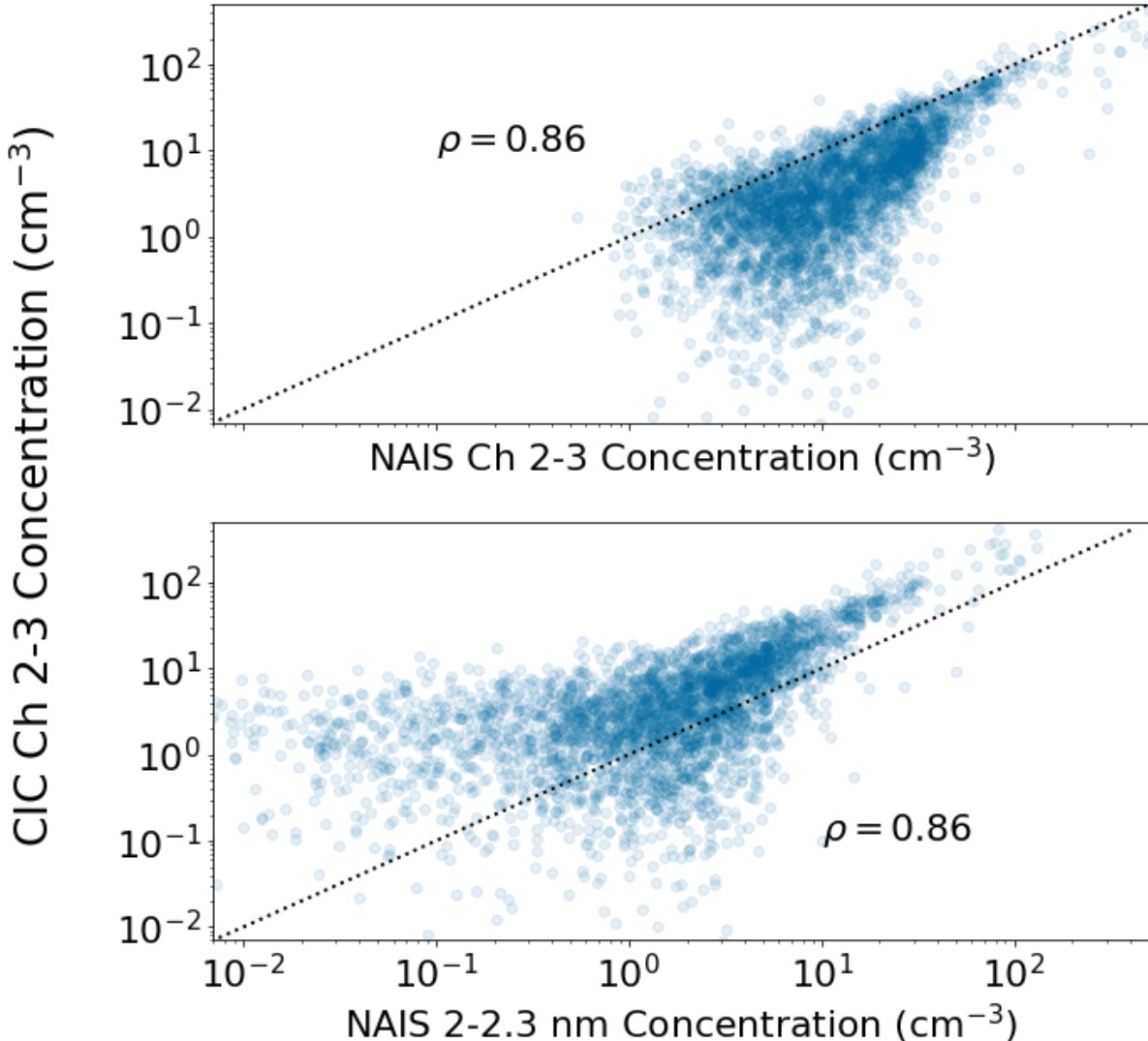

Figure 3. Scatter plot of approximately 2.0–2.3 nm negative ion 15-minute-median
concentrations measured with the CIC as a function of concentrations measured with the
NAIS in Hyytiälä. The NAIS concentrations on the top figure were determined for the same
size range as covered by the CIC Channels 2 and 3 (for details, see Sect. 2.3). The NAIS
concentrations on the bottom figure are for the diameter range 2.0–2.3 nm. The black dotted
line marks the 1:1 line. Pearson correlation coefficient ρ of the two concentrations shown is
included in the figure.

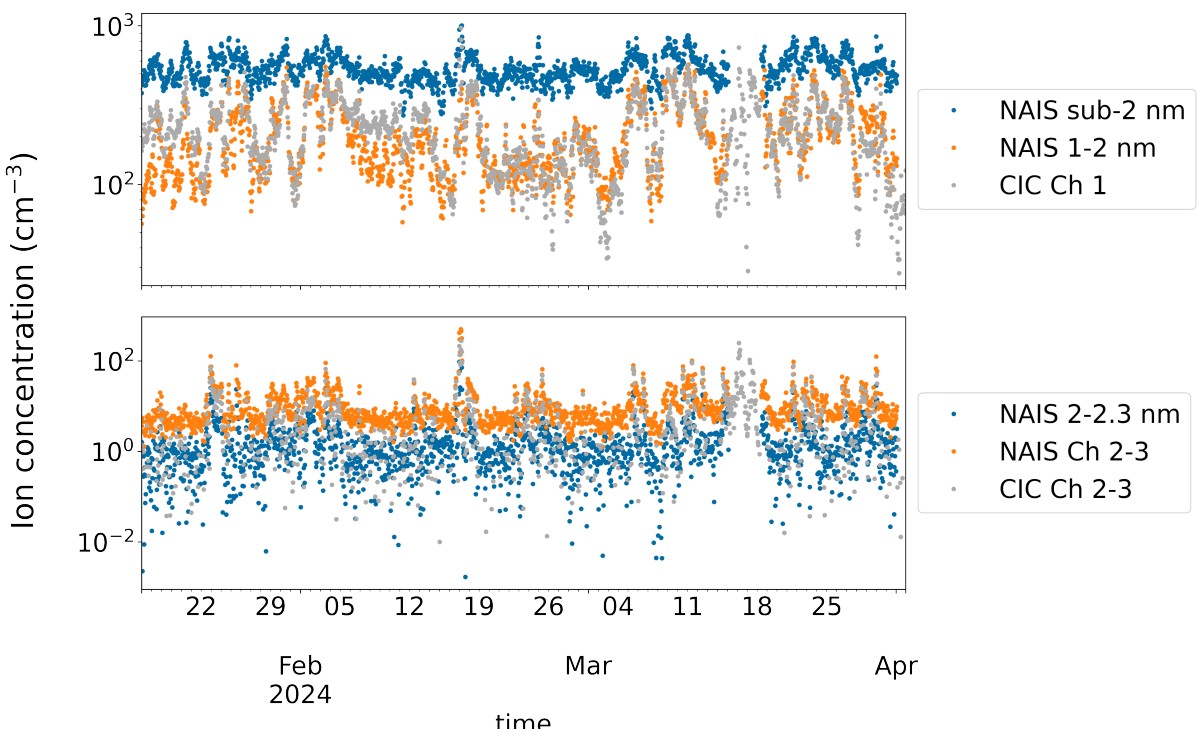

Figure 4. Time series of observed ion concentrations. The top figure has the concentrations of
small ions from the CIC Channel 1 and from the NAIS for both all sub-2 nm ions and 1–2 nm
ions. The bottom figure has concentrations of ions measured by the CIC channel 2–3 which
approximately corresponds to the size range of 2.0–2.3 nm. In addition, there are
concentrations of 2.0–2.3 nm ions measured by the NAIS (NAIS 2.0–2.3 nm) and
concentrations from the NAIS that were determined for the exact same size range as covered
by the difference of CIC Channels 2 and 3 (NAIS Ch 2-3).

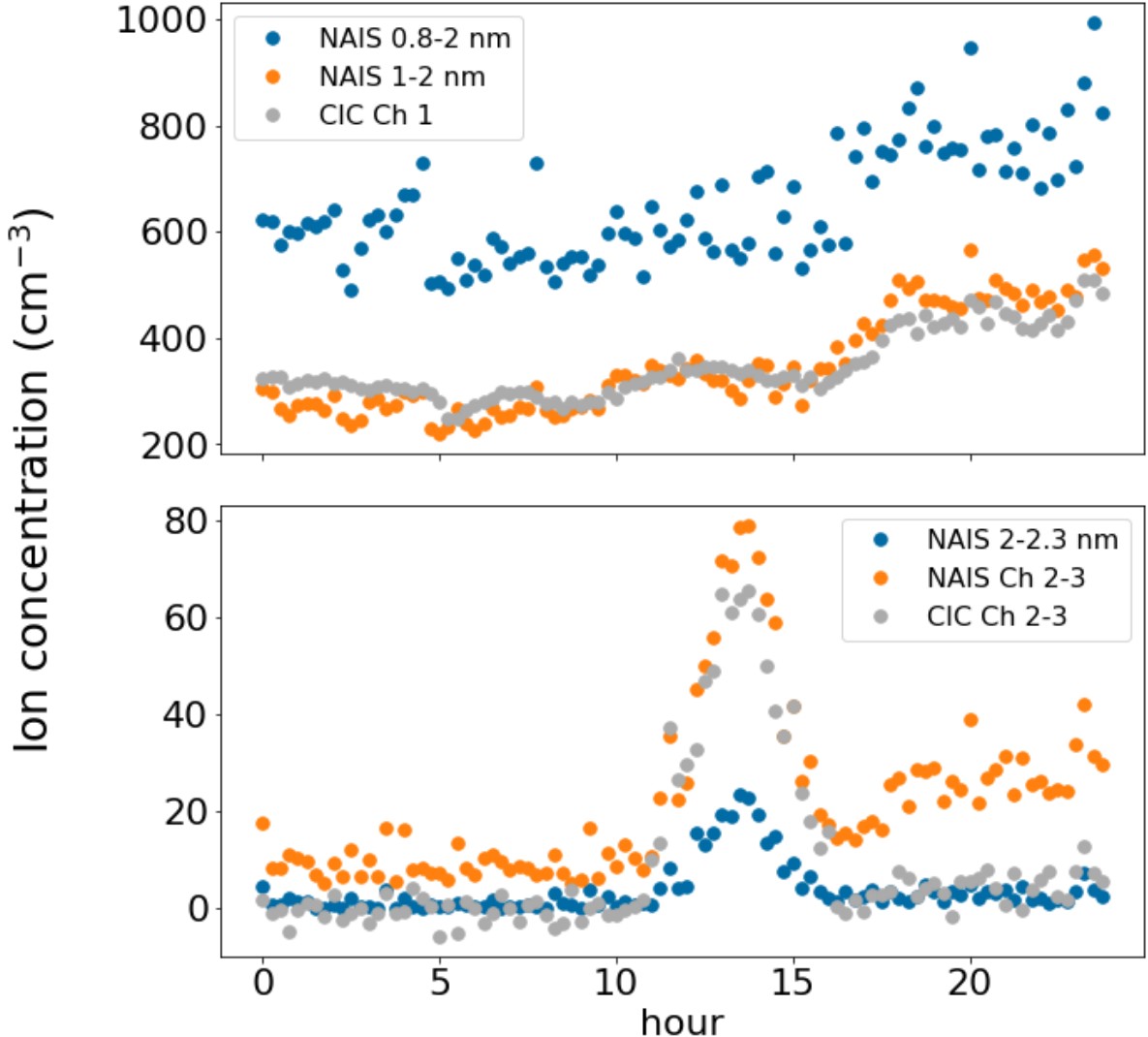

Figure 5. Observed negative ion concentrations on 10.03.2024. The top figure has the
concentrations of small ions. For CIC, they are from the CIC Channel 1. From the NAIS,
concentrations for all measured sub-2 nm ions and based on the size range 1–2 nm are
included.  The bottom figure has the concentrations of intermediate ions. For CIC, they are
from Channel 2–3, corresponding to roughly 2.0–2.3 nm size range. For NAIS, the
concentrations of ions between 2.0 and 2.3 nm are included, as well as the concentrations that
were determined for the exact same size range as covered by the CIC Channels 2 and 3
(NAIS Ch 2–3). The correlation coefficients on this day are 0.83, 0.95, 0.93 and 0.90 for
NAIS 0.8-2 nm vs CIC Channel 1, NAIS 1–2 nm vs CIC Channel 1, NAIS 2.0-2.3 vs CIC
Channel 2–3, and NAIS Channel 2–3 vs CIC Channel 2–3, respectively.

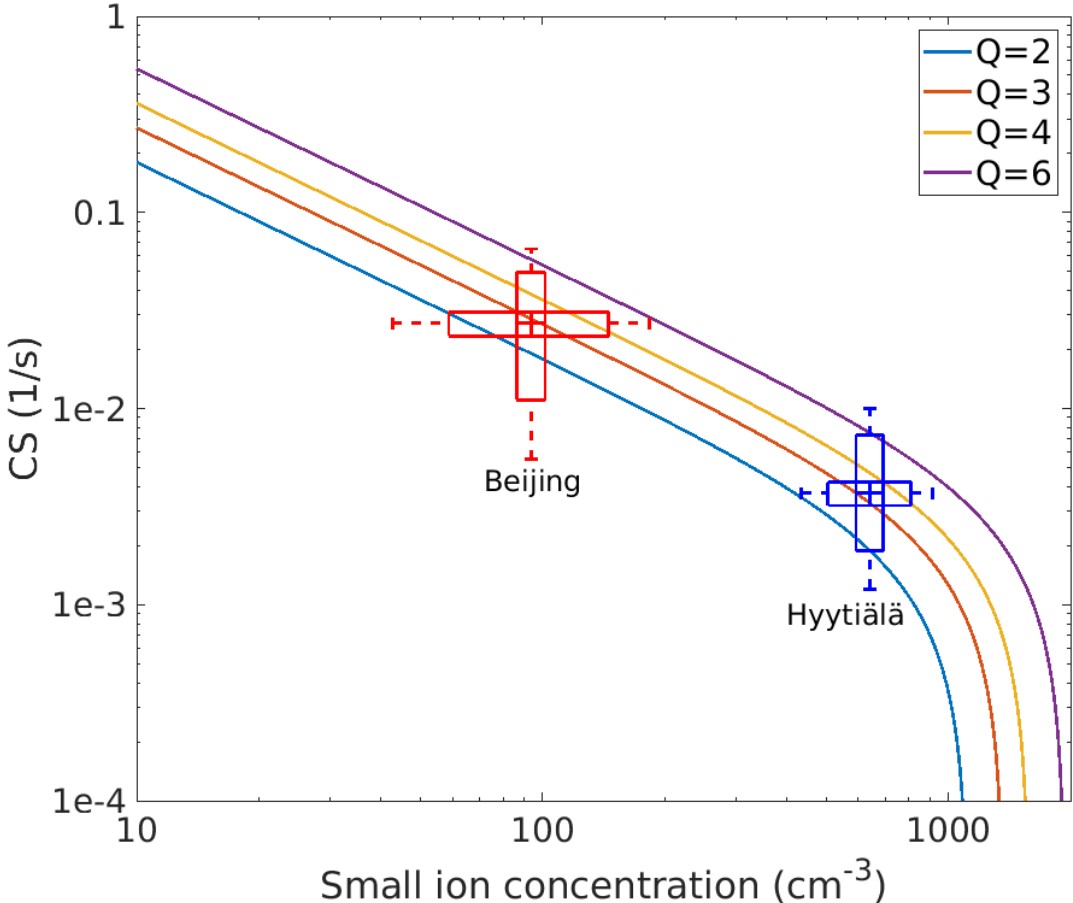

Figure 6. Condensation sink (CS) as function of the small ion concentration for different ion
source rates ($Q$, ions cm$^{-3}$ s$^{-1}$). The observed values of $I$ and CS in Hyytiälä and Beijing
(medians marked by the center line of the boxplot, 25% and 75% quartiles marked by the
edges, and 10% and 90% percentiles marked by the whiskers of the boxplots) indicate ion
source rates between about 2 and 4 cm$^{-3}$ s$^{-1}$ in both places.

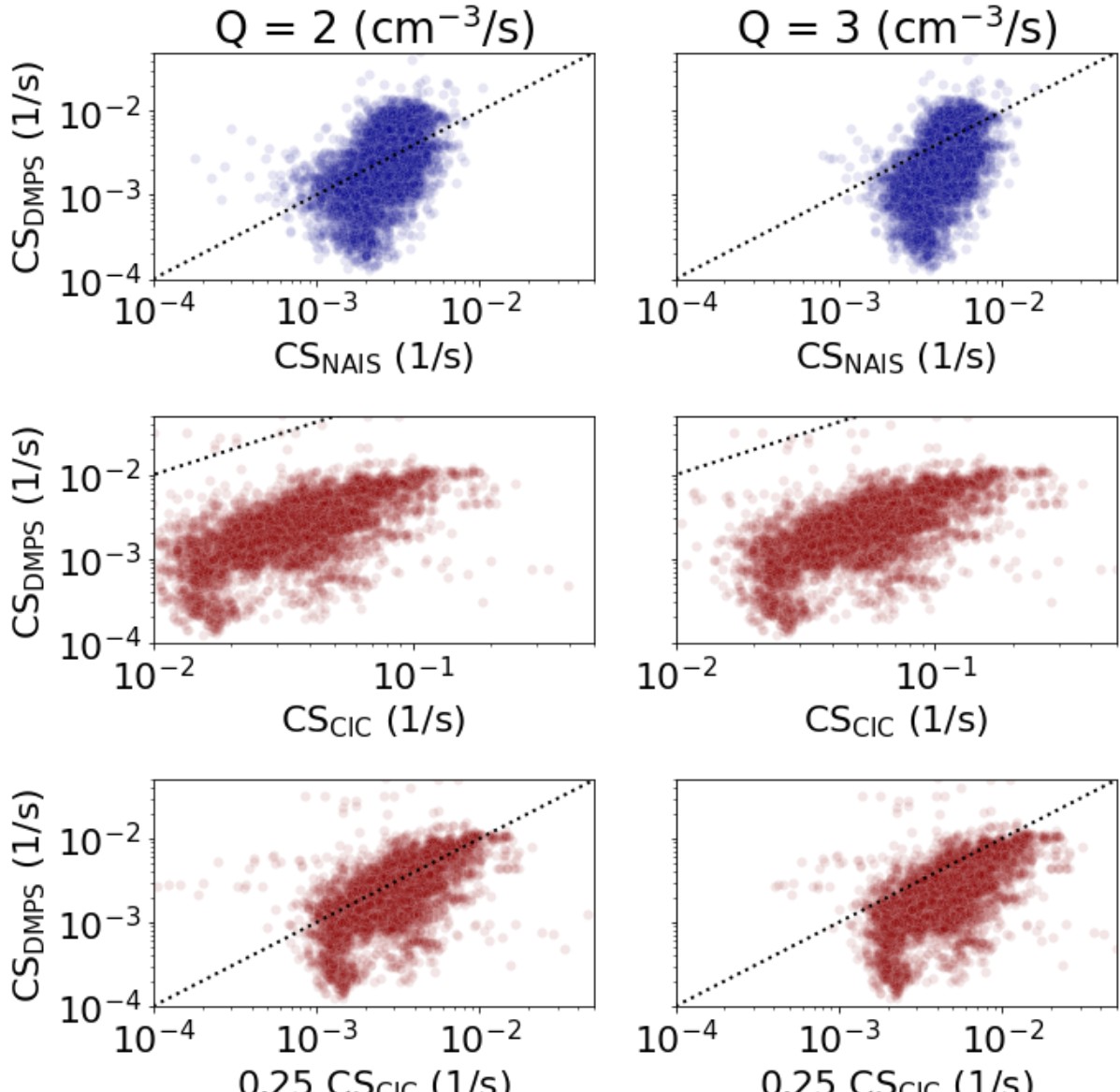

Figure 7: Condensation sink (CS) determined based on particle number size distribution data measured by DMPS versus CS derived based on negative sub-2 nm ion concentrations from NAIS and CIC. For CIC and NAIS, Eq. 5a and 5b have been used, respectively.

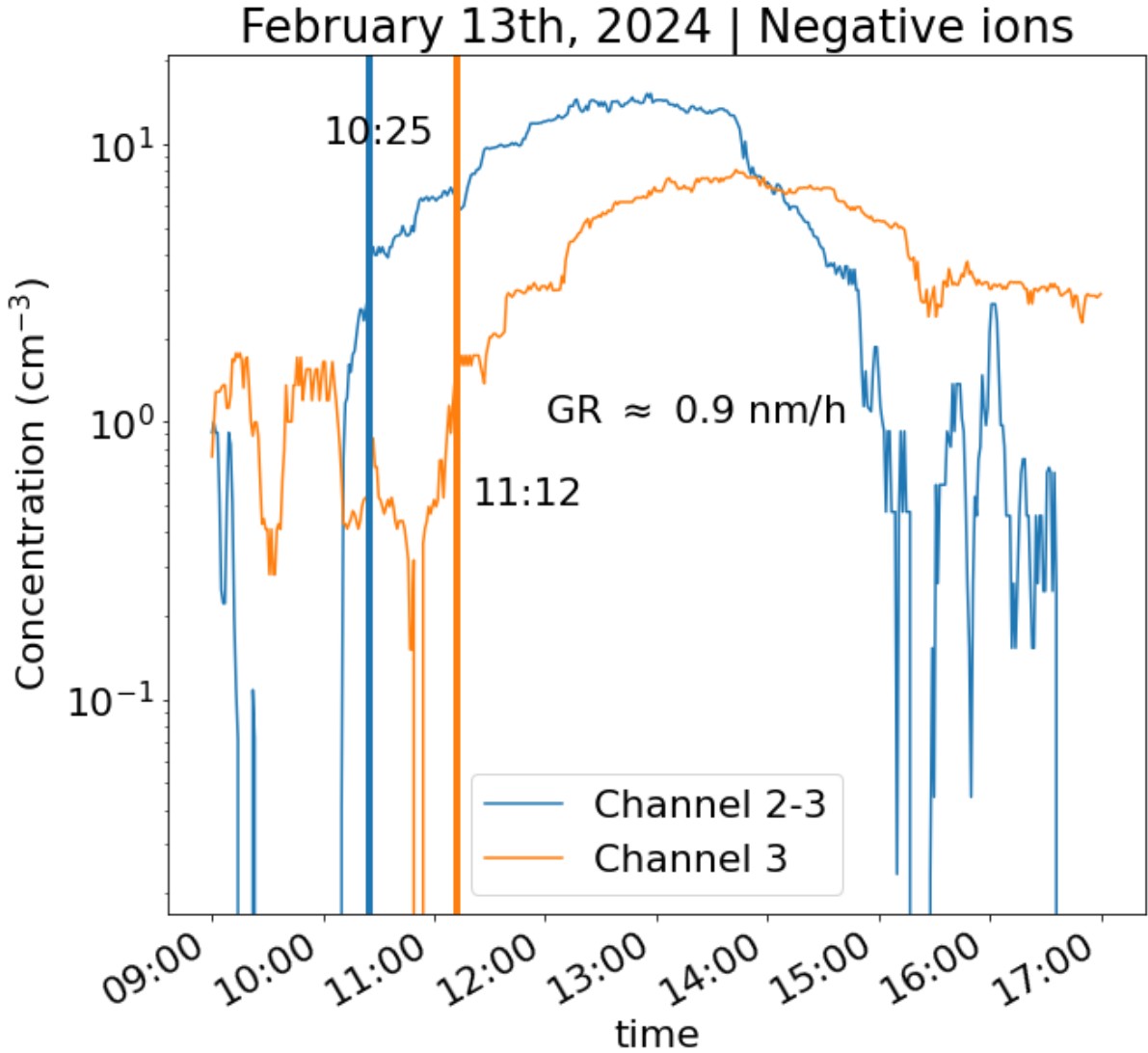

Figure 8: The CIC Channel 3 and Channel 2-3 concentrations on the day of February 19th.
Approximate appearance times have been marked by vertical lines alongside the growth rate
(GR) from 2.2 to 2.9 nm derived based on those appearance times.

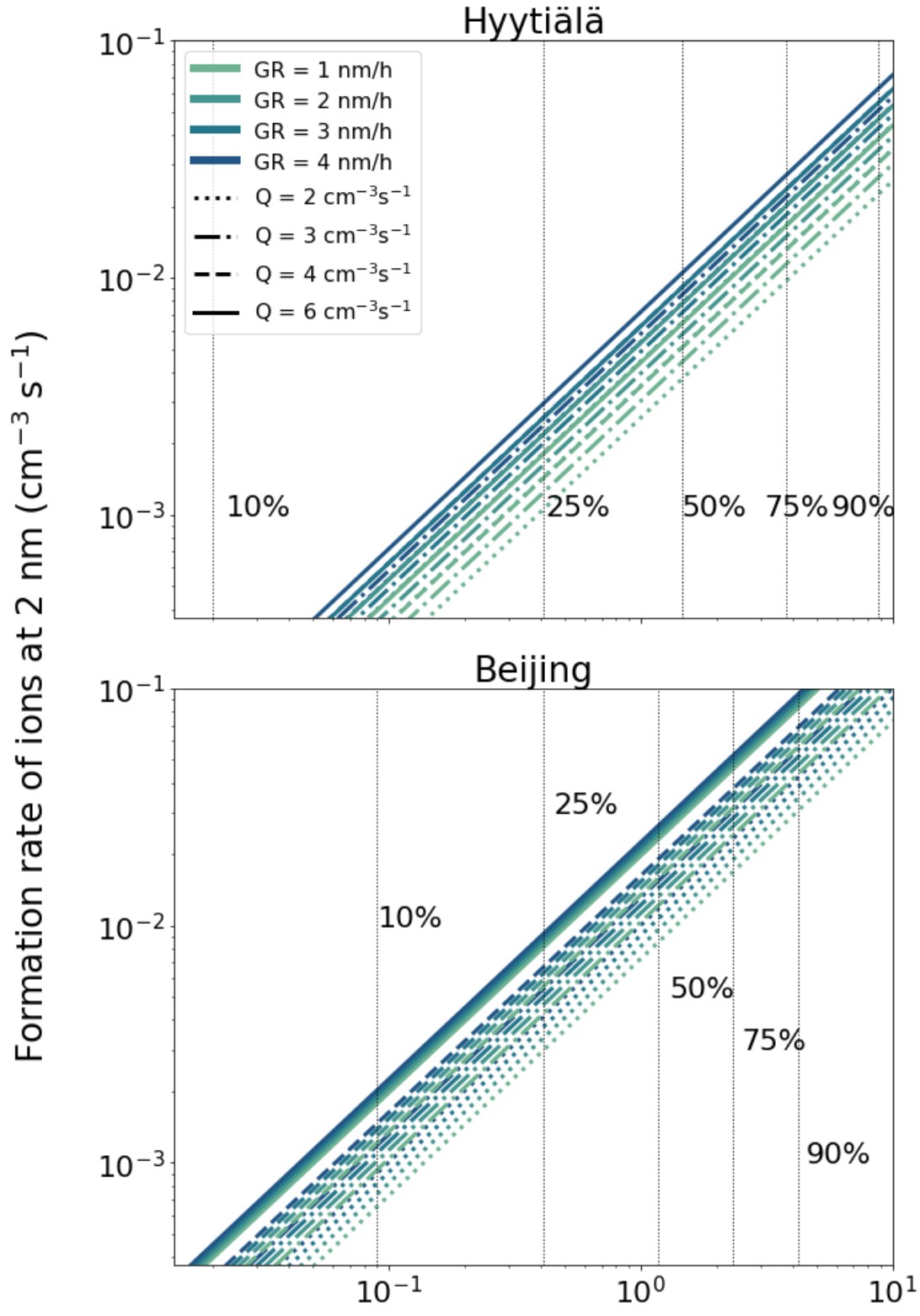

Figure 9: The estimated formation rate of 2 nm negative ions as a function of the
concentration of 2.0-2.3 nm ions. The ion growth rate has been assumed to be equal to 1
nm/h.The 10%, 25%, 50%, 75, 90% concentration values are indicated by the vertical lines.

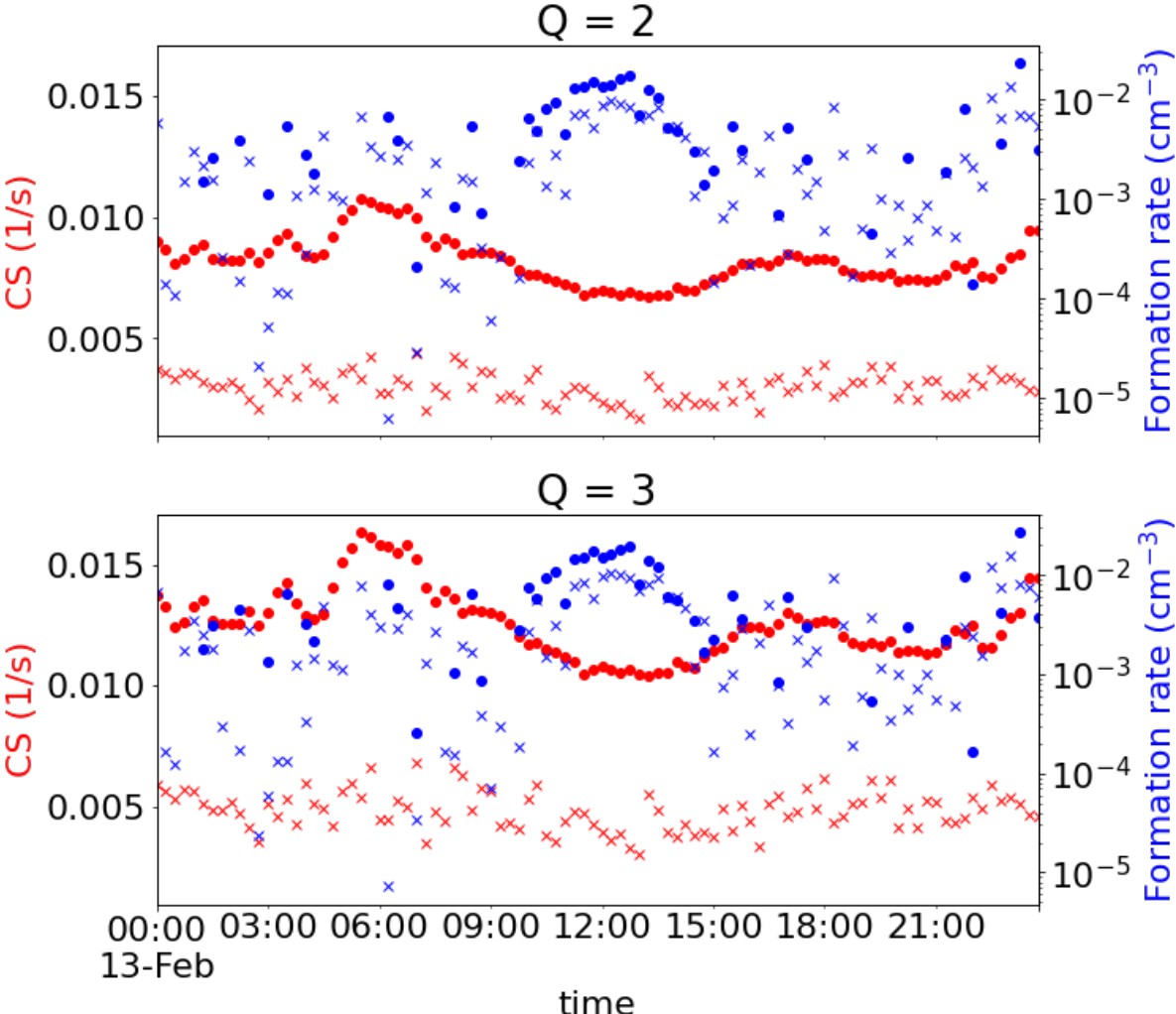

Figure 10: Condensation sink (right) and formation rate of 2 nm ions (left). The values marked by dots are based on CIC channel 1 and channel 2-3 ion negative ion concentrations while the values marked by x markers are based on NAIS sub-2 nm and 2.0-2.3 nm negative ion concentrations. The top panel has valued with assumed ion source rate of Q= 2 cm$^{-3}$ s$^{-1}$ while the bottom panel includes those for Q=3 cm$^{-3}$ s$^{-1}$ . A value of 0.9 nm/h for GR used, as determined in Fig. 7 for this day. Negative and positive ion concentrations were assumed to be the same.