# Peer review of "On the potential of Cluster Ion Counter (CIC) to observe local new particle formation, condensation sink and growth rate of newly formed particles"

_Aerosol Research, 2024_

## Author Comment (AC1)

RC1: ['Comment on ar-2024-14'](), Anonymous Referee #1, 28 Jun 2024

The MS is based on a really neat idea, and having a not too sophisticated instrument to detect / trace ion formation and its role in new particle formation events at a local level would be an important step forward. The MS therefore is certainly within the scope of Aerosol Research and could be a highly valuable contribution to the field. There are some issues with the MS itself, however, that should be fixed before it can be accepted for publication.

**We thank the reviewer for positive and constructive comments. Our responses to these comments are given separately after each comment in bold. All line numbers in our answers correspond to the revised version of the manuscript.**

The MS presents data obtained with a novel instrument, but as this instrument (the Cluster Ion Counter, CIC) is not described in the MS, it is impossible to put the results into context. The reference describing the CIC (Mirme et al., 2024) is given as "to be submitted", and this is definitely insufficient. Of course a "modified CIC" is used for this study, but without info on the original, the modifications, which are described in the MS, cannot be adequately appreciated. I therefore strongly suggest the authors add a dedicated section on the CIC, its operation and expected improvements of the modified CIC to the current MS.

**This is a very valuable comment and we have expand the first paragraphs of section 2.1:**

**"The Cluster Ion Counter (CIC) is an instrument for measuring the total number concentration of both positive and negative cluster ions. The CIC uses two separate first-order cylindrical differential mobility analyzers, one for each polarity (Tammet, 1970). The principal components of the analyzers are a central electrode on the axis of the analyzer that is held at a steady voltage, and three cylindrical collecting electrodes flush with the outer wall of the analyzer which are at zero electric potential. A constant sample flow is produced through the analyzer using a blower at the outlet. The sampled ions passing through the analyzers are repelled by the central electrode and they may deposit on one of the collecting electrodes depending on the electrical mobility of the ions. The electric current produced by the deposited ions is measured using high precision integrating electrometers (Mirme et al., 2024).**

**The mobility dependent detection efficiency curves of the three channels are determined by the geometry of the analyzer, sample air flow rate and the electric voltage of the central electrode. According to the idealized model of differential mobility analyzers (Tammet, 1970), the primary parameters governing the detection efficiency curves and the limiting mobilities of the collecting electrodes are the electrical capacitances between the central electrode and the each collecting electrode, as well as the ratio of sample flow rate to central electrode voltage. The original CIC was designed to allow the estimation of average cluster ion mobility. However, the device can easily be modified to focus on other aspects of the mobility distribution.**

**In the CIC, the flow rate to voltage ratio can be freely adjusted through software. The lengths of the collecting electrodes and geometry of the central electrode of the CIC can be changed without requiring additional modifications to the device."**

**REF:**

**Tammet, H.: The aspiration method for the Determination of Atmospheric-Ion Spectra, The Israel Program for Scientific Translations Jerusalem, National Science Foundation, Washington, D.C., 1970.**

**Below, a figure (to be published in Mirme et al., 2024 Design and performance of the Cluster Ion Counter (CIC)) illustrating the structure of the instrument is shown.**

[Figure]

**Figure: The CIC has two identical cylidrical analyzers to simultaneously measure positive and negative ions. Each analyzer has a central electrode on the axis that is held at a steady voltage and three collecting electrodes on the outer wall, each detecting different ranges of mobility diameters. The analyzers are preceded by switchable high-voltage inlet filters to allow automated zero level current measurement.**

The efficiency curves shown in Figure 1 should be explained in more detail. Channel 1 can be used to estimate the total ion concentration in the whole size range. But as the efficiency of CH 1 is way higher than the efficiency of CH 2 and CH 3 even at the sizes where their respective efficiencies peak, and there never is mention of subtracting counts from CH 2 and CH 3 from the CH 1 counts, the values given for "small ions" (i.e. < 2nm) seem to be much overestimated – or was there some extra data processing not mentioned in the MS? Clarification of this issue is definitely needed.

**The small ions concentration estimate derived from Channel 1 could be improved by subtracting the signal of either Channel 2 or 3 multiplied by an appropriate coefficient to correct for the contribution of > 2.0 nm particles and achieve a virtual sharper cut-off size. However, in practice, small ion concentrations detected by Channel 1 are at least an order of magnitude higher than those of larger ions detected by Channels 2 or 3. So the benefit would be negligible while the measurement error would be increased.**

Lines 210 - 211: "… overall agreement between these two instruments is very good…." Looking at the figures, the good correlation coefficient does not really suggest a "very good" agreement. The correlation coefficients are ok, but the data deviate strongly from the 1:1 line with the concentrations in CIC CH 1 higher than those obtained by the NAIS (Figure 2). In both plots shown in Figure 3 a line "drawn by eye" also shows quite a large deviation from the 1:1 line. This issue is exacerbated by the fact that all these plots are log-log plots, so it should be discussed. The huge discrepancy between NAIS and CIC data in Figure 5, lower panel, should also be discussed in more detail.

**We have changed the phrasing from "very good" to a more moderate "good". In addition, we have added more discussion on these Figures (Lines 263-273):**

**"We can see that when the small ion concentration is above 200 cm⁻³, the two instruments show similar values while at smaller concentrations there is more spread in the values with CIC generally measuring higher concentrations than NAIS. At low concentrations, it is possible that the uncertainties in the detection efficiencies of the ions with diameters close to 1 nm impact the results, explaining our observations. CIC Channel 2-3 concentration are consistently lower than NAIS Channel 2-3 concentrations, with the difference being smaller when the concentrations are higher, suggesting that a lower concentrations noise impacts the comparison increasingly. There is more spread between the values of NAIS 2.0-2.3 nm and CIC Channel 2-3. At higher concentrations the CIC shows higher concentrations than NAIS 2.0-2.3 nm concentration."**

Discrepancies between 2.0-2.3 nm ions from NAIS and CIC are addressed in lines 284-286: "CIC Channel 2-3 is likely influenced by ions larger than 2.3 nm, impacting the measured concentration when the intermediate ion concentration is high, such as during NPF"

Lines 291-295: "... Figure 4 and 5 show that the low background concentrations measured by CIC Channel 2-3 are, on average, less than 10% of NAIS Channel 2-3 concentrations, which we postulate is due to estimation errors caused by the limited size resolution of the NAIS as well as different background noise levels of the instruments. At very low concentrations, the values from either instrument can be considered unreliable. "

In view of the lacking info about the CIC, the issue with the efficiency curves and the vast scatter of data in Figures 2 – 5, the conclusion statement in lines 286 – 288 seems to be over-optimistic ("are we able to utilise a simple ion counter to find out LIIF in a proper way. According to our results presented above, the answer is: yes"). The statement at the end of the conclusion section, however, can only be underlined: "if we want to investigate aerosol formation and growth rates for the nucleation mode (3 – 25 nm) ..... NAIS measurements are needed"

**We have now addressed the influence of discrepancies in the background concentrations of 2.0-2.3 nm ion concentrations on our results in the revised manuscript (Lines 296-301)**

**"Regardless, within the scope of this study, these background concentrations are of less interest compared to the higher concentrations. Periods of LIIF can be identified based on elevated 2.0-2.3 nm ion concentrations, and these ion concentrations can then be used to derive quantities, such as the ion formation rate, to determine the intensity of LIIF. The comparison of the two instruments done here has shown that we can use CIC measurements to identify LIIF."**

**We have also clarified the connection of this paper to LIIF, and reformulated the main objectives (Lines 80-81): " ... formation rate of 2 nm ions, quantifying the intensity of local new particle formation (actually local intermediate ion formation, LIIF), based on CIC measurements."**

**Lines 402-403: "are we able to utilise a simple ion counter to identify and quantify LIIF in a proper way?"**

**Lines 408-411: "Furthermore, the CIC makes it possible to estimate the growth rate from about 2 nm to 3 nm and, with this information, the formation rate of 2-nm ions, which we can use to quantify the intensity of LIIF."**

**Hopefully with these additions, the conclusions now seem less overly optimistic.**

Other points:

The structure of the MS could be improved – it does not make sense mentioning Figures 6 – 9 first and discuss Figures 2 ff afterwards. The section on observation data might be shifted to a later position in the MS

**The structure has been improved by taking out the unnecessary reference to Figures 6-9.**

Lines 200 – 205: no mention is made on the influence of the efficiency of the CIC Channel 1 – discuss the effect

**We have now addressed how detection efficiency impacts the discrepancy between CIC and NAIS sub-2 nm concentration (Lines 253-256): "This result is expected, as the detection efficiency of both instruments decreases rapidly for particles smaller than 1 nm. However, the NAIS is able to correct for this in data inversion, while the CIC is not due to the lack of detailed information about the measured size distribution."**

Line 243 "NAIS are 0.237, **258** and 0.266 times ...." Missing "0." In front of "258"....

**Corrected.**

Lines 273 – 275: discuss reason for the vastly different values for the formation rates of 2 nm particles at the different measurement locations

**We added the following sentence at the end of this paragraph (Lines 385-390):**

**"These results are fully consistent with the general finding that on average, observed new particle formation rates are 1 to 3 orders of magnitude larger in polluted urban environments compared with clean or moderately polluted environments (Kerminen et al., 2018; Nieminen et al., 2018), whereas the average formation rates of 2 nm ions are typically within a factor of 2-3 between different environments (Manninen et al., 2010)."**

Statements lacking references:

line 46 "much less information is available" – unless the authors mean "no info", references should be added here.

**We added two references here: Kulmala et al., 2024a, 2024b.**

**Kulmala, M., Ke, P., Lintunen, A., Peräkylä, O., Lohtander, A., Tuovinen, S., Lampilahti, J., Kolari, P., Schiestl-Aalto, P., Kokkonen, T., Nieminen, T., Dada, L., Ylivinkka, I., Petäjä, T., Bäck, J., Lohila, A., Heimsch, L., Ezhova, E., and Kerminen, V.-M.: A novel concept for assessing the potential of different boreal ecosystems to mitigate climate change (CarbonSink+ Potential). Boreal Env. Res., 29, 1–16, 2024a.**

**Kulmala, M., Aliaga, D., Tuovinen, S., Cai, R., Junninen, H., Yan, C., Bianchi, F., Cheng, Y., Ding, A., Worsnop, D. R., Petäjä, T., Lehtipalo, K., Paasonen, P., and Kerminen, V.-M. (2024) Opinion: A paradigm shift in investigating the general characteristics of atmospheric new particle formation using field observations, Aerosol Res., 2, 49-58, https://doi.org/10.5194/ar-2-49-2024, 2024b.**

line 130: basis for approximation "d_p,i ... equal .... 1.2 nm for negative cluster ions...."

**We have now mentioned this in the revised manuscript (Lines 169-171) : "The $d_{p,I}$ were determined as weighted mean diameters of 0.8-2.0 nm (NAIS) and 1.0-2.0 nm (CIC) negative ions based on the NAIS ion number size distributions. The concentrations of ions in different size bins were used as weights."**

**RC2:** 'Comment on ar-2024-14', Anonymous Referee #2, 03 Jul 2024

Review of "*On the potential of Cluster Ion Counter (CIC) to observe local new particle formation, condensation sink and growth rate of newly formed particles*" by Markku Kulmala, Santeri Tuovinen, Sander Mirme, Paap Koemets, Lauri Ahonen, Yongchun Liu, Heikki Junninen, Tuukka Petäjä and Veli-Matti Kerminen for consideration for publication in **Aerosol Research**

*General Comments*

In this manuscript, the performance of a modified Cluster Ion Counter (CIC) to accurately measure the small ion and intermediate ion concentration is evaluated. The Authors use a Neutral cluster and Air Ion Spectrometer (NAIS) as a benchmark instrument to validate the CIC, for a period of several weeks of measurements, in the winter and spring of 2024. The Authors also develop a set of equations that allows the derivation from CIC measurement, of important new particle formation (NPF) characteristics such as the condensation sink, the formation rate of intermediate ions, and their growth rate. They then go on to apply these equations on an example day with observed NPF: February 13[th], 2024.

This study fits well within the scope of Aerosol Research. The use of the CIC to characterize localized and regional NPF may well be important for NPF research, but the Authors fail to clearly articulate how, why or in which cases the CIC measurements are advantageous over NAIS measurements, and build a convincing narrative to support their conclusions. Some linkages between the results presented and their consequences are left unsaid, leaving the reader to fill in the blanks. Some sections and statements seem out-of-place and distract from the scientific narrative the reader is after. The different parts of the manuscript are often not logically connected and different vocabulary is used in different sections. While I do believe this research is deserving of eventual publication, it is also deserving of a better presentation and introduction to the Aerosol Research readership.

**We thank the reviewer for positive and constructive comments. Our responses to these comments are given separately after each comment in bold.**

*Specific comments*

Abstract

The abstract should include a statement as to how this research is contributing to advancing the field. What can we do with these tools that couldn't be done before? What is this research enabling?

**We added the following sentence (Lines 35-38): "The new CIC is a simple and cheap instrument that can be used in different environments to obtain information about small ion dynamics, LFII and CS in a robust way when combined with the theoretical framework presented here**."

1. Introduction

Last paragraph (lines 68-80): While the previous paragraph closes stating that the NAIS is essentially too good an instrument for the purposes, this last paragraph of the introduction does not state how or in which circumstance the CIC is a better choice than the NAIS. Is it because it is cheaper? Is it because less maintenance is required?

**Yes, CIC is cheaper and needs less maintenance. We have added a sentence into conclusions (Lines 424-426): "Since the CIC is ca seven times cheaper and requires less maintenance than NAIS, with CIC one can get data more numerous observation locations and have a wider data coverage than with NAIS."**

Lines 74 and 75 (as well as line 28 in the abstract): the Authors mention that they will derive an equation to estimate the "intensity" of local NPF or intensity of local new particle formation or local intermediate ion formation, LIIF. This is the last time the word "intensity" appears in the manuscript. Similarly, LIIF is not discussed anywhere in the core of the manuscript, until it comes back in the Conclusions section. It is not clear which equation represents the "intensity" of NPF or LIIF. Assumedly, the Authors mean $J_2$. Please harmonize the words used to describe different quantities throughout the manuscript.

**We have clarified this in the revised manuscript:**

**Rephrasing (lines 78-81): "In order to reach these objectives, we will first derive equations that can be used to estimate condensation sink (CS), growth rate of newly formed particles and formation rate of 2 nm ions, quantifying the intensity of local new particle formation (actually local intermediate ion formation, LIIF), based on CIC measurements."**

**Adding (lines 297-301): "Periods of LIIF can be identified based on elevated 2.0-2.3 nm ion concentrations, and these ion concentrations can then be used to derive quantities, such as the ion formation rate, to determine the intensity of LIIF. The comparison of the two instruments done here has shown that we can use CIC measurements to identify LIIF."**

**In the conclusions, rephrasing (lines 402-403): "The recent progress on finding local NPF (e.g. Kulmala et al., 2024; Tuovinen et al., 2024) has opened a question: are we able to utilise a simple ion counter to identify and quantify LIIF in a proper way?"**

**And lines 408-411: "Furthermore, the CIC makes it possible to estimate the growth rate from about 2 nm to 3 nm and, with this information, the formation rate of 2 nm ions, which we can use to quantify the intensity of LIIF."**

2.1 Cluster Ion Counter (CIC)

Lines 91-98: In this section, we learn that the Authors are using a modified CIC, where the analyzer has been modified. First, please define what an analyzer is, or show it in a schematic of the modified CIC. Second, please explain how the analyzer was modified. If it is described elsewhere, please reference to this description, for example, at the end of the first sentence, on line 95.

**The CIC description has been modified (see the answers to REF 1.)**

Figure 1: Please name the x-axis "mobility equivalent diameter", or specify which kind of diameter is being measured. Also consider making the legend more explicit by identifying which ion sizes each of these lines represent (except for channel 2).

**The axis has been renamed.**

Figure 1: In the main text, please explain the criteria for deciding the channels' boundary diameters.

**We have expanded on this in the revised manuscript (Lines 127-132):**

**"We altered the collecting and central electrode geometry, as well as voltage, and flow rate within the mechanical constraints of the original device so that the transfer functions of channel 2 and 3 would differ only in a relative narrow size range and the difference would peak between 2.0 and 2.3 nm. This**

**required extending the first collecting electrode and shortening the second and third electrode, as well as changing the diameter and length of the central electrode."**

**This new paragraph, alongside the expanded explanations on the CIC in the previous answers, should hopefully clarify the issue.**

2.2 Conceptual Model

Please introduce what this is with a sentence or two at the beginning of the section. Also "conceptual model" is a rather vague name for a specific model or set of equations. Consider giving it a more specific name and adjusting the references in the text accordingly.

**We replaced "conceptual model" with "Theoretical framework".**

Line 111: please correct the units for the ion-ion recombination rate, $\alpha$.

**Corrected.**

Equation 4: improve the presentation of this equation, as possible, so that it is easier to read.

**We rearranged and modified the terms of this equation to make it more readable.**

Lines 129-131: How were these $d_{p,1}$ values chosen? Based on what curves, is it somewhere in the literature?

**We have added a clarification (Lines 169-172): "The $d_{p,I}$ were determined as weighted mean diameters of 0.8-2.0 nm (NAIS) and 1.0-2.0 nm (CIC) negative ions based on the NAIS ion number size distributions. The concentrations of ions in different size bins were used as weights."**

Equations 5a and 5b: indicate which is for the CIC and which is for the NAIS next to the equation rather than using parentheses.

**Corrected.**

Line 156: Perhaps it would be worth discussing the validity of this assumption, or referring to a previous discussion of this assumption.

**We modified the text (Line 195): "Assuming a pseudo-steady state ($dN/dt = 0$), ...". The validity of the pseudo-steady state assumption of any quantity depends on the speed at which its sinks and source terms change in time. This assumption is expected to be reasonable for ion concentrations for most of the time, but may fail in extreme conditions.**

2.3 Observations

This entire paragraph appears out of place and references to Figures 6, 8 and 9 are anachronic. I think the Authors are trying to describe the data available for each measurement station. If the Authors want to specify which data was used for each Figure, this could be presented in an orderly manner in a Table among Supplementary Materials.

**Corrected by removing references to figures 6-9.**

With the sentences about Figs 6, 8, and 9 out of the way, this paragraph is still hard to follow and would benefit from being reorganized in a more logical manner.

**We agree and this is corrected by removing ref to figs 6-9**

3.1 Instrument Comparison

It is clear that the NAIS is considered the instrument of reference here, with good reasons. However, the Authors should point to justificative materials in the literature, such as an evaluation of the NAIS' accuracy, or it's predominance in the concerned Research Communities.

**Since with the CIC we want to look at ion dynamics, ion spectrometers are the only instruments that CIC should be compared with. The NAIS is an instrument that is widely used, and it has also been compared with other instruments, e.g. in CLOUD experiments.**

Table 1 & Table 2 captions: "Statistics", please replace by which statistics, assumedly "Percentiles"?

**Corrected.**

Table 1: "the negative concentrations [...] are indicative of a noisy signal of the instrument" The implications of this should be discussed in the main text as well as the transformations applied (noting that there are no negative numbers in Fig. 3). Based on Table 1, it looks like a significant proportion of the concentration data is negative so how these data were handled and why are important. Plotting these numbers (including the negative ones) against the Virtual CIC channel 2-3 might be useful in telling where the CIC channel 2-3 becomes unreliable.

**Figure 3 has logarithmic scales, and therefore negative values are not visible. No other transformations have been applied. The logarithmic scaling has been chosen over the linear one due to the resulting figure being more clear.**

**In general, at such low concentrations the signal from either instrument can be considered quite unreliable. In addition, within the scope of this study, the lower values of 2.0-2.3 nm ions are not of interest. We have added the following discussion to address this (Lines 289-301): "Comparing the lower percentiles in Tables 1 and 2, it is apparent that a large fraction of CIC Channel 2-3 concentrations are negative. At very low concentrations (< 1 cm$^{-3}$), the signal is mainly noise. In addition, Figures 4 and 5 show that the low background concentrations measured by CIC Channel 2-3 are on average less than 10% of NAIS Channel 2-3 concentrations, which we postulate is due to estimation errors caused by the limited size resolution of the NAIS as well as different background noise levels of the instruments At very low concentrations, the values from either instrument can be considered unreliable. Regardless, within the scope of this study, these background concentrations are of less interest compared to the higher concentrations. Periods of LIIF can be identified based on elevated 2.0-2.3 nm ion concentrations, and these ion concentrations can then be used to derive parameters, such as the ion formation rate, to quantify the intensity of LIIF. The comparison of the two instruments done here has shown that we can use CIC measurements to identify LIIF."**

Table 2 caption (lines 352-355): This explanation of the virtual channel, which in my opinion is name awkwardly, needs to be presented in the main text, named, and its name be used consistently across the manuscript. The description of how it was calculated does not belong in the caption. See my related comments under Figure 3.

**The description has now been removed from the caption and is included in the Methods (Lines 218-227).**

Line 207: I think the word "detailed" can be taken out, or replaced by a more appropriate qualifier. Figures 2 and 3 rather appear to be bulk comparisons.

**The word "detailed" has been removed.**

Line 209: Even though "larger" here refers to the concentration, the Authors will understand that the phrase "somewhat larger small ions" is not ideal. To be consistent with the second part of the sentence, which speaks of "lower concentrations", I suggest the Authors use the word "higher small ions [...] concentration".

**Due to the changes made to the paragraph, this sentence has been replaced and reads (Lines 263-266): "We can see that when the small ion concentration is above 200 cm$^{-3}$, the two instruments show similar values, while at smaller concentrations there is more spread in the values with the CIC generally measuring higher concentrations than the NAIS"**

Figure captions in general: there are a lot of explanations in the captions that are being repeated in many captions. That information should be moved to the main text and does not need to be included in the captions. For example, once the measurement period has been defined in section 2.3, there is no need to point to the dates where measurement happened and did not happen at every figure caption. This information is unnecessarily repeated in Figs. 2, 3 and 4.

**The figure captions have been shortened and unnecessary information removed from then, and instead appears in the main text.**

Figures 2 and 3: I would like to see the density of these points. A way to show that would be to box the points and show boxplots, or fit an unconstrained linear fit, or add transparency to the points. All we see at the moment is a mostly fully coloured area.

**We have added transparency to the points.**

Lines 207-212: Nowhere in this paragraph is the fact that Figure 3 has two panes discussed. Nowhere is the difference between those two panes explained. This needs to be done in the main text, around this paragraph.

**Further discussion has been added and both panels have now been mentioned.**

Figure 3: What is the difference between the calculation of NAIS Ch 2-3 and NAIS 2-2.3 nm? Which one of those two is equivalent to the "Virtual CIC channel 2-3" presented in Table 2? Since the CIC channel 2-3 is considered to cover the 2-2.3 nm range, how are these two concentrations different? This is not clear from the caption and these calculations should anyway be described in the main text. I also see that the upper pane is supposed to be covering a "wider size range of ions". What is that range exactly?

**We have clarified these two different concentrations in the Methods. Virtual CIC channel 2-3 has been replaced with Channel 2-3 to avoid confusion. As can be seen from Figure 1, CIC Channel 2-3 does cover a slightly wider diameter range than 2.0-2.3 nm, and as such CIC Channel 2-3 gives only an approximate value for the concentration of 2.0-2.3 nm ions. To consider the impact of this potentially wider range of diameters of measured ions, Virtual Channel 2-3 concentrations were determined from NAIS data.**

**This has now been mentioned in the revised manuscript (Lines 218-221): "In addition, as CIC Channel 2-3 covers a slightly wider diameter range than 2.0-2.3 nm, we determined concentrations corresponding to those within the same mobility diameter range from the ion number size distributions measured by NAIS (NAIS Channel 2-3)."**

Figure 3 caption (lines 392-396): This describes a calibration of the NAIS data, using the CIC as a reference. Firstly, this procedure should be described and defined in the main text (and be removed from the caption). Secondly, it is highly unusual to calibrate the reference instrument with the instrument under test. Unless there is a very good reason to present it this way, I strongly recommend flipping the roles so that the CIC

concentration is calibrated to the NAIS concentration. This being said, the reasons behind this unusual manoeuvre may become clearer once the difference between the upper and the lower pane is clarified.

**We have clarified the issue in the revised manuscript by adding more details on what was actually done (Lines 221-227): "The NAIS ion number size distributions were multiplied by the detection efficiencies for the CIC Channel 2–3 (Figure 1), and then summed. The resulting total concentrations were assumed to correspond to the detected ion concentration by CIC Channel 2-3. This concentration was then divided by the average detection efficiency for the CIC Channel 2-3 to get the atmospheric ion concentration. If the NAIS concentrations are assumed to be equal to the atmospheric concentrations, then in theory the CIC and NAIS Channel 2-3 concentrations should be equal."**

Figure 4 caption: same comments as for Figs. 2 and 3. Trim down the caption and have these descriptions of calculations once in the main text. Use their names consistently throughout the manuscript. Same confusion as to what the difference between NAIS Ch 2-3 and NAIS 2-2.3 nm is.

**Corrected.**

Figure 5: The legends should be inside the figure or the proportions be changed: they take way too much space at the moment.

**Corrected.**

Figure 5 caption: Much of the caption is repeated information. Please clean it up.

**Corrected.**

Figure 5 implications: In the text, please discuss why the CIC agrees better with NAIS Ch 2-3 at high concentration but better with NAIS 2-2.3 nm at lower concentrations. Was this behaviour expected?

**We have briefly addressed this in the text now; see previous comments for the explanation.**

Line 216: Why was March 10$^{th}$, 2024 selected? How does it compare to other days?

**From those days, during which there was a clear peak in the intermediate ion concentrations, March 10$^{th}$, 2024, was selected at random. As a large fraction of the data is from the winter season, many of the days had consistently low ion intermediate concentrations and March 10$^{th}$ has higher daytime concentrations than the average day during the measurement period.**

3.2 Application of CIC measurement in investigating condensation sink and local NPF

Figure 6 caption (line 437): Small ions are defined as $I$ in the text, not N. I assume it should be $I$. Also, the last 3 lines should be removed as it is already explained in Section 2.3.

**Corrected.**

Line 225: Is this Eq. 5a or 5b? Which instrument are we assuming? Section 2.3 states that the CS was determined using a PSD system.

**The Eq. 5 used here is 5b as the ion concentrations used are based on NAIS data for negative total sub-2 nm ions. It is now also mentioned in the text.**

**We have added the following clarifications to this (Lines 306-308): "In the same plot, we have included the observed variability of CS as determined from the particle number size distributions and $I$ in both Hyytiälä and Beijing."**

Line 243: Those numbers provided have 3 digits after the decimal. Are they all significant digits? Also, since the NAIS is the reference instruments, consider flipping the ratio to have the CIC CS be around 4 times those measured by the NAIS. Also, this, and Tables 1 and 2, sort of contradict your statement on lines 232-233, that the CIC has a higher detection efficiency, unless the NAIS data is inverted and the CIC data isn't. How do you explain the difference? In section 2.3, the Authors state that a DMPS and PSD system were used to determine the "CS data". Where is that shown or used? Which CS is closer to the truth? How applicable is this ratio between CIC-CS and NAIS-CS across days and environments? I think there should be a more in-depth discussion of how the CS differs for CIC and NAIS measurements. After all, it is one of the main products of this manuscript.

**Using 3 digits we can see that the overall estimation 0.25 is justified.**

Figure 7 caption (lines 459-465): This discussion should be in the main text. Furthermore, the limitations of this technique to estimate the CS should be discussed.

**We have moved most of the discussion from the captions to the main text.**

**Our estimates for CS based on CIC measurements are within a factor of three compared to CS calculated from DMPS data, which is now stated in lines 331-332: "Assuming Q=2, the CS values predicted by CIC are mainly within a factor of three from $CS_{DMPS}$ values. "**

**The limitations of this technique to estimate CS in general are also discussed on lines 334-340: "We have assumed that the only losses of ions are due to their coagulation with larger particles and their recombination with oppositely charged ions. In reality, processes such as deposition also affect the ion concentration. For example, Tammet et al. (2006) found that in Hyytiälä deposition of ions to forest canopy impacts small ion concentrations. In addition, we have assumed the ion source rate to be constant. In reality, it is expected to vary somewhat, for example due to varying radon concentration (e.g., Hirsikko et al., 2007). Therefore, the presented method of determining CS can only give a rough approximation for CS."**

**REF: Hirsikko, A., Paatero, J., Hatakka, J., and Kulmala, M.: The [222]Rn activity concentration, external radiation dose and air ion production rates in a boreal forest in Finland between March 2000 and June 2006, Boreal Environ. Res., 12, 265–278, 2007.**

**Tammet, H., Hõrrak, U., Laakso, L., and Kulmala, M.: Factors of air ion balance in a coniferous forest according to measurements in Hyytiälä, Finland, Atmos. Chem. Phys., 6, 3377–3390, doi:10.5194/acp-6-3377-2006, 2006.**

Figure 7: The vertical lines should be thicker to distinguish them from spikes in the time series. The second vertical line should also be orange to match its associated time series.

**Corrected.**

Figure 7: Please add a subplot or a figure in the Supplementary Material where the appearance time method is used to determine the GR from NAIS data on the same day. Compare the GR obtained with the CIC to the one obtained with the NAIS.

**Unfortunately, it is challenging to determine the GR from NAIS ion number size distribution, and no reliable results could be obtained for the example day. The figure attached below illustrates the poor linear fit to the appearance times, from which GR from NAIS data was calculated. Therefore, we have chosen not to include it in the manuscript.**

[Figure]

Line 252: 2.1 nm is used here without explaining what it represents. Is 2.1 nm considered to be the peak of the Ch2-Ch3 detection efficiency curve? Eyeballing it, I would use a different number myself, hence the importance of being clear about where it comes from. A 0.1 nm error on a range of 0.8 nm could make a difference in the GR it gives.

**We have changed 2.1 nm to 2.2 nm based on the detection efficiencies and their peaks from Figure 1, changing GR from approximately 1 nm/h to 0.9 nm/h.**

Lines 250 and around: please refer to Eq. 10, or if it is too simple, remove it from Section 2.2.

**Added.**

Lines 259-260: This statement suggests that it is only possible to determine the GR under certain circumstances. What are these limitations, is it GR-related? Is it determined by the NPF's intensity? What is the percentage of days where the GR is retrievable? Have you tried it for other days?

**This was addressed in lines 361-366: "We should note, however, that it is not possible to determine GRs for all measurement days using the procedure presented here. This is because even if an increase in ion concentrations was observed, the signal might be too noisy, making the determination of appearance times too unreliable. In addition, not all days exhibited a clear delay between the two appearance times, making the determination of growth rate impossible."**

**During the period from which the used data is from, these days from which GR can be retrieved are in the minority. We expect that the fraction of days from which GR can be determined increases with an increasing intensity of NPF.**

Lines 262-275: It is not clear whether any measurement data is used to generate the J2. Looks like they are look-up figures where a potential user would determine the 2.0-2.3nm concentration using the CIC and

then plug that in the equation or look at Figure 8 or 9 to find the corresponding J2. Is that correct? Either way, this paragraph needs to be clearer.

**We have added the following sentence to this paragraph to clarify (Lines 375-376): "For Figure 9, the median sub-2 nm ion concentrations in Hyytiälä and in Beijing were used in Eq. 9."**

**The following sentence has also been modified for clarity (Lines 370-373) : " ...formation rate can be given as a function of the measured number concentrations of 2.0–2.3 nm intermediate ions, in addition to which $J_2$ depends on the growth rate, ion source rate, and ion loss rate, the latter of which was estimated using the sub-2 nm ion concentrations according to Eq. 5b."**

Lines 273-275: while true and important, this statement appears unrelated to the rest of the contents in the paragraph. A simple sentence modification would probably explain why the is mentioned here. For example, "If one wanted to estimate the total formation rate...", assuming this is the reason this sentence is here for.

**Corrected as proposed.**

Conclusions and summary

In general, I don't feel that the conclusions are supported by the material in the manuscript. The conclusions feel detached from the manuscript, although connected to the introduction (only).

**We have improved the connection between results and conclusions.**

Line 287-288: "find out LIIF in a proper way". I'm not sure what that means. Please rephrase. Also, since we haven't seen any mention of LIIF since the introduction, linking LIIF to the contents of the manuscript would be essential.

**We modified the text in the following way (Lines 402-403): "... are we able to utilize a simple ion counter to identify and quantify LIIF in a proper way?"**

Line 290: "a somewhat modified version of the CIC" Take out "somewhat" and add the reference to line 94. That said, Mirme et al., 2024 doesn't seem to be available yet. Some short explanation of what this modification entails would be appropriate in Section 2.1.

**Added**

Line 292: What is LIIF? What are its units? Where in the manuscript was it estimated?

**LIIF refers to local intermediate ion formation. The answer to the second comment to the Introduction should help clarify the issue.**

Line 294: "ion sinks" is not mentioned anywhere in the manuscript. Did you mean Coagulation sinks or ion-ion recombination, or the sum of both? Avoid introducing new vocabulary and concept in the conclusions section.

**We mean "coagulation sink of ions". Corrected.**

Line 296-298: Is the portability of these equations and measurement principles to positive ions theoretical or have the Authors actually measured and evaluated it? The Authors suggest in the last sentence of this paragraph, that it might not actually work for positive ions. I suggest toning down the claims that the same is valid for positive ions.

**Theoretical principles are the same for both negative and positive ions. However, the diameters of positive sub-2 nm ions are larger than of negative ones, impacting Eqs. 5a and5 b. The impact should be small to moderate. In addition, while our results suggest that CS estimated from CIC is 4 times that of estimated by NAIS for negative ions, exactly the same ratio is not expected to be true for positive ions.**

Line 300: Please, make clear that the "estimated" values are based on CIC measurements. Also, I have seen ion concentrations compared, but I have not seen a direct comparison of CS between the CIC and any other instrument.

Line 301: I have not seen evidence that CIC measurements were used in Beijing. Please clarify.

**Answer to the 2 comments above: We modified this sentence into the following form: "The comparison of the estimated condensation sink from ion concentrations using the ion balance equation with the observed ones in Hyytiälä and Beijing demonstrates how the CIC.."**

Line 302: "ion sinks" is mentioned again. It is not defined. Consider defining the term around Line 111 or mention its components instead.

**Again, this refers to the coagulation sink of ions. Corrected**

Line 305: I'm still not sure what LIIF is and how the CIC is effective to observe it. Which Figures show this?

**Previous answers should hopefully clarify the first part of the question**

**We have addressed the latter part of the question (lines 299-301): "Periods of LIIF can be identified based on elevated 2.0-2.3 nm ion concentrations, and these ion concentrations can then be used to derive quantities, such as he ion formation rate, to determine the intensity of LIIF. The comparison of the two instruments done here has shown that we can use CIC measurements to identify LIIF."**

*Typos, etc.*

Throughout the manuscript: Consider using 2.0-2.3 nm instead of 2-2.3 nm so both boundaries have the same assumed accuracy.

**This is a valid suggestion, corrected.**

Table 1: Please fix the alignment of the titles so they align with their respective columns. Also add 'small ions' and '2-2.3 nm' under the current titles for clarity.

**Corrected.**

Line 215: no "s" to ion.

**Corrected.**

Line 217: "pretty well": that is a rather vague qualifier. Consider editing.

Lines 218-220: "captured consistently"? Please rephrase this. Probable meaning: peaks coincide?

**This paragraph has been modified to be more exact and now it reads (Lines 279-287): "Total sub-2 nm ion concentrations measured by the NAIS are higher than CIC Channel 1 ion concentrations. However, for**

**majority of the time (see Figure 4), the NAIS 1-2 nm ion concentration and CIC Channel 1 concentration are close to each other. On the selected day, CIC Channel 2-3 peak NAIS Channel 2-3 values are similar, 60 and 80 cm$^{-3}$, respectively, whereas the NAIS 2.0-2.3 nm peak value is lower at around 20 cm$^{-3}$. CIC Channel 2-3 is likely influenced by ions larger than 2.3 nm, impacting the measured concentration when intermediate ion concentration is high, such as during NPF. The correlation coefficient between the concentrations from the two instruments on the selected day is around 0.9 for both sub-2 nm and 2.0-2.3 nm ions."**

Line 234: dependent on particle size (remove "a")

**Corrected.**

Line 243: assuming 256 was meant to be 0.256?

**Corrected.**

Line 450: Suggested: Determining the ion growth rate (GR) based on CIC measurements.

Line 455: "similar to appearance time method" Is it similar to or is it the appearance method? Either remove "similar to" or describe how this method differs from the Lehtipalo et al. appearance time method.

**Answer to the 2 above questions: we have rewritten the relevant parts:**

**(Figure 8, caption): "Approximate appearance times have been marked by vertical lines alongside the growth rate (GR) from 2.2 to 2.9 nm derived based on those appearance times."**

**More details given in lines 345-354: "The ion concentrations were smoothed using a moving 1-hour median method to lessen the impact of noise. As we can see from Figure 8, Channel 3 and Channel 2–3 concentrations on the selected day have a similar shape between 10:00 and 16:00, and the shape of the Channel 3 roughly follows that of Channel 2–3 with a time delay. Considering the shape and features of the two curves, and the times at which the two concentrations reach a similar fraction of the maximum concentration (appearance time method), two time instances were identified visually. The appearance times were chosen to correspond to times when the ion concentrations were around 20 % of the maximum concentrations. From these approximate appearance times, a time delay was calculated."**

Line 457: consider using 20% instead of 0.2.

**Corrected.**

Line 458: consider replacing "gained" with calculated or identified.

**We wrote "calculated".**

Line 253: "can be considered a very realistic one" à do you mean that the value fits well within the expected range?

**We have reformulated this as suggested.**

Figures 8 & 9: consider merging these figures into one figure with two subplots.

**Corrected.**

Figure 8, y-axis: Formation rate "of negative ions" at 2 nm.

**Corrected.**

Line 300: The name of the internationally famous Hyytiälä station is misspelled!

**Corrected.**

Line 304: "pretty well" is pretty vague. Please quantify how well they agree.

**This sentence has been modified and now reads: "We also compared the CIC with the NAIS in Hyytiälä, which demonstrates that the measured ion concentrations from CIC are able to capture the temporal behavior of the ions such as the variation in concentrations due to LIIF."**

- **RC3**: 'Reply on RC2', Anonymous Referee #2, 03 Jul 2024

Clarification regarding my comment on line 300 (I am RC2): Figure 10 shows a comparison of CS for one day as a time series, but I would like to see a CS vs CS figure spanning the whole campaign to be convinced of the claims made in the conclusion.

**We have added the below figure and the following discussion to the revised manuscript.**

[Figure]

**Figure 7: Condensation sink (CS) determined based on particle number size distribution data measured by DMPS versus CS derived based on negative sub-2 nm ion concentrations from NAIS and CIC. For CIC and NAIS, Eq. 5a and 5b have been used, respectively.**

Lines 327-332: "Figure 7 shows the CS derived based on Eq. 5a and 5b versus CS determined from the full particle number size distribution ($CS_{DMPS}$). We see that the CS predicted by the NAIS varies less than the $CS_{DMPS}$, but is mostly within the same order of magnitude. The CS predicted by the CIC is consistently higher than the $CS_{DMPS}$. However, considering the above discussion, and multiplying the estimated CS by 0.25, we get values much closer to $CS_{DMPS}$. Assuming Q=2, the CS values predicted by the CIC are mainly within a factor of three from $CS_{DMPS}$ values."

RC4: 'Comment on ar-2024-14', Anonymous Referee #3, 08 Jul 2024

This is an interesting analysis of the performance of the cluster ion counter (CIC). Overall, the paper is well written, and the results generally support the overall view that this instrument, while simple, can give some insights into important aspects relevant to atmospheric nucleation. I recommend acceptance but I would like the authors to consider the following comments, which I feel are needed to improve the paper:

**We thank the reviewer for positive and constructive comments. Our responses to these comments are given separately after each comment in bold.**

Line 45: I am not sure how "averaged" applies to regional npf ... certainly we can average anything over large spatial scales. I recommend replacing with "that takes place"

**Corrected.**

Line 65: I think the reader would like to know what, specifically, is the unnecessary information that NAIS provides.

**The CIC is cheaper and easier to maintain, and therefore it is a proper instrument for small ion dynamics, LIIF and CS estimations. With the NAIS we can look in more details growth rates in the size range 2 nm to 40 nm. However, also with the NAIS we do not get exact CS, and the detailed knowledge from other channels are not needed for CS estimation and LIIF.**

**We have added to this sentence (Lines 68-71): "The NAIS is, however, a sophisticated instrument that provides information not necessarily needed when investigating local NPF, such as detailed knowledge of both ion and particle number size distributions."**

Line 127: Does the fact that ions are charged have any impact on their condensation sink rate? It appears that only the physical size of ions is considered here, but in other applications (such as ion-neutral reactions) charge does matter. Please discuss.

**Condensation sink is a property of molecular compounds (those that can condense onto clusters or aerosol particles, whereas for both neutral particles and ions their coagulation sink is the relevant property. To avoid misunderstandings, we have added before eq. 2 that this equation hold for neutral particles. With this addition, eqs. 2 and 3 lead to equation 4 in a straightforward way, and there should be no ambiguities how CS is related to the (coagulation) sinks of ions and neutral particles.**

Lines 302 and 304: The phrases "relatively accurate" and "pretty well" are not helpful. Please be more specific based on your results.

**The parts in question has been rewritten with more precise language in mind (Lines 415-423): "We compared the CIC with the NAIS in Hyytiälä, which demonstrates that the measured ion concentrations from CIC are able to capture the temporal behavior of the ions such as the variation in concentrations due to LIIF. The comparison of the estimated condensation sink from ion concentrations using the ion balance equation with the observed ones in Hyytiälä and Beijing demonstrates how the CIC, together with the simple theoretical framework, can be used to estimate condensation sink, coagulation sink of ions and the ion formation rate. In addition, the comparison of estimated CS based on CIC measurements with the CS determined particle number size distributions shows that we can get estimates that are within a factor three of the real CS"**

Minor comment:

Line 288: If this is a question, the last phrase should end with a question mark. The rule: "If each point is a complete sentence, capitalize the first word and end the sentence with appropriate ending punctuation."

**Yes this is. We added a question mark.**